



# Influence of cloud microphysics schemes on weather model predictions of heavy precipitation

Gregor Köcher[1], Tobias Zinner[1], and Christoph Knote[1,2]

[1]Meteorologisches Institut, Ludwig-Maximilians-Universität, Munich, Germany
[2]Medizinische Fakultät, Universität Augsburg, Augsburg, Germany

**Correspondence:** Gregor Köcher (gregor.koecher@physik.uni-muenchen.de)

**Abstract.** Cloud microphysics is one of the major sources of uncertainty in numerical weather prediction models. In this work, the ability of a numerical weather prediction model to correctly predict high-impact weather events, i.e., hail and heavy rain, using different cloud microphysics schemes is evaluated statistically. Polarimetric C-band radar observations over 30 convection days are used as observation dataset. Simulations are made using the regional-scale Weather Research and Forecasting Model (WRF) with five microphysical schemes of varying complexity (double moment, spectral bin (SBM), and particle property prediction (P3)). Statistical characteristics of heavy rain and hail events of varying intensities are compared between simulations and observations. All simulations, regardless of the microphysical scheme, predict heavy rain events that cover larger average areas than those observed by radar. The frequency of these heavy rain events is similar to radar-measured heavy rain events, but still scatters by a factor of 2 around the observations, depending on the microphysical scheme. The model is generally unable to simulate extreme hail events with reflectivity thresholds of 55 dBZ and higher, although they have been observed by radar during the evaluation period. For slightly weaker hail/graupel events, only the P3 model is able to reproduce the observed statistics. Analysis of the raindrop size distribution in combination with the model mixing ratio shows that the P3, Thompson 2-mom, and Thompson aerosol-aware models produce large raindrops too frequently, and the SBM model misses large rain and graupel particles.

## 1 Introduction

High-impact weather events, e.g. heavy rain or hail, can lead to massive economic losses and threaten lives and livelihoods. The severe flood event in July 2021 in western Germany and neighboring countries, for example, resulted in the death of at least 170 people and insured losses of more than 10 billion euros in Germany alone (Junghänel et al., 2021; Schäfer et al., 2021). To prepare population and infrastructure and thereby reduce these losses, national weather services typically use numerical weather prediction (NWP) models to forecast such extreme events with lead times of several days. At the German Weather Service (DWD) the ICON (Zängl et al., 2014) model is used for this purpose. However, the accuracy of NWP forecasts is limited, and some hazardous weather events, such as small-scale convective events, are difficult to predict in terms of correct location, timing, or magnitude. For example, in Belgium, operational forecast models were unable to predict a thunderstorm that produced a strong downburst less than 100 m in diameter and caused 5 fatalities (De Meutter et al., 2015). There is poten-





tial for improvement in several respects: increasing the resolution (e.g., Clark et al., 2016; Morrison et al., 2015), allows more and more processes to be simulated explicitly and effectively eliminates problems caused by inaccurate parameterizations. Another aspect with potential for improvement in NWP models is the treatment of cloud microphysics (e.g., Morrison et al., 2015; Rajeevan et al., 2010), which will be the focus of this study. Microphysical processes occur on very small scales and are typically parameterized. Modelling these processes is challenging and subject to large uncertainties (Morrison et al., 2020; Fan et al., 2017). A variety of different microphysical schemes are used in current operational NWP models, varying greatly

in their complexity. Typically, these schemes are categorized as either "bulk" or "bin" schemes, although other categories exist too. Briefly, "bulk" schemes predict one or more moments of a predefined statistical function representing the droplet size spectrum, while "bin" schemes predict mass and number concentrations for a range of sizes ("bins") without imposing a particular size distribution. A good overview of the differences, advantages, and disadvantages of bulk and bin schemes can be found in

Khain et al. (2015). Depending on the choice of microphysical scheme, simulation results and required computational power can vary greatly. Very simple microphysical schemes (e.g., Kessler, 1969) are cheap, but also do not capture the complexity of real microphysical processes. As the complexity of the microphysics scheme increases, so does the computational power required, which is often the limiting resource in weather prediction. Therefore, an important question to answer is: How much complexity in microphysical schemes is required for numerical weather prediction? Microphysical models often make very

simple assumptions, e.g. some hydrometeors are usually simply assumed to be spherical. In part, this is due to a lack of knowledge: It is not known exactly which processes are poorly represented in numerical weather prediction models (Morrison et al., 2020). One reason is that direct observation of microphysical processes is very difficult due to the size of the particles involved at millimeter scales and below, as well as the many different shapes, sizes, or phases of the hydrometeors involved. Therefore, in order to constrain the processes implemented in an NWP model, observations are needed that provide just this information.

Direct (in-situ) measurements in clouds, for example with aircraft, provide this information, but such measurements are expensive and require much effort. Moreover, it is not possible to cover several parts of the cloud or several clouds at the same time; in situ measurements are spatially limited. Radar observations, in contrast, provide measurements over a large atmospheric volume with high temporal and spatial resolution. However, the processes are not measured directly, but inferred from the measured reflectivity. This is associated with numerous uncertainties, e.g., due to beam broadening effects, non-uniform beam

filling, attenuation along the beam path, or variation of the refractive index due to different particle phases (e.g., hail or rain) in the same measurement volume. In addition, the reflectivity strongly depends on the particle size distribution: larger particles usually dominate the reflectance signal, since the (Rayleigh) back scattering (or radar cross section) is proportional to the sixth power of the droplet diameter (Bringi and Chandrasekar, 2001).

In recent years, polarimetric radar measurements have become available. In 2009, DWD began upgrading the national radar

network to full dual-polarimetric radars that perform operational polarimetric volume scans over all of Germany with a temporal resolution of 5 minutes (Helmert et al., 2014). This data is highly useful because the polarimetric information is affected by many particle population properties, such as particle phase, density, orientation, shape, size, and number concentration, and thus can be used to evaluate and improve NWP models (Ryzhkov et al., 2020). Such a dataset provides striking opportunities to evaluate properties and abilities of cloud microphysical schemes in NWP models which were previously inaccessible. In prin-





ciple, NWP model output can be compared to polarimetric radar signals in two different ways: (1) converting the model output
to polarimetric radar signatures using a polarimetric radar forward operator (e.g., Ryzhkov et al., 2011; Augros et al., 2015;
Snyder et al., 2017) or (2) retrieving microphysical information from the polarimetric radar signal (e.g., Cao et al., 2010). These
two approaches and related literature are described in a review paper by Ryzhkov et al. (2020). Polarimetric information can be
used for many different applications, such as improved quantitative precipitation estimation (QPE) from radar measurements

(Ryzhkov et al., 2022), hydrometeor identification (HID) algorithms (e.g., Park et al., 2009), or microphysical ice retrievals
(e.g., Tetoni et al., 2022). The focus of this study is on using polarimetric radar signals to evaluate cloud microphysics schemes
of a NWP model. For this assessment, we use both of the aforementioned approaches: (1) by applying a radar forward oper-
ator, we generate simulated polarimetric radar signals, and (2) by applying a hydrometeor classification, we obtain dominant
hydrometeor classes from the observed radar signals, which we then compare to the simulated hydrometeors.

There are numerous publications that evaluate weather model predictions and microphysical schemes for specific events of
interest that were particularly hazardous, well observed, or both (e.g., Shrestha et al., 2022; Taufour et al., 2018). This is an
important approach to understand the behavior of a model in specific scenarios. However, atmospheric conditions are highly
variable, and a model may perform very well in a particular situation but may provide very poor predictions under different
atmospheric conditions. Statistical analysis of weather forecasts over a longer time period and across multiple weather events

provides a more robust assessment of model performance, but requires a large amount of effort because a large number of
weather simulations and observations must be performed and checked for quality. Depending on the grid spacing and type of
model used, the available computational power may also simply be insufficient to provide weather simulations in a limited
amount of time. In a recent study by Köcher et al. (2022), microphysics schemes are statistically evaluated over a dataset of
30 convection days in 2019 and 2020. This dataset consists of weather simulations with 5 different microphysics schemes of

80  varying complexity, as well as simultaneous polarimetric C-band radar observations from DWD, and is therefore well suited
to address the aforementioned following problems:

1. How can polarimetric radar observations be used to statistically evaluate microphysics schemes?

2. What complexity of microphysics schemes is required for NWP predictions?

In Köcher et al. (2022), cloud microphysics of varying complexity are assessed by a statistical comparison of the observed

85  radar signals with the simulated radar signals from the model output, i.e., by applying the approach (1) described above. This
study builds on the study by Köcher et al. (2022) and goes one step further by additionally using the approach (2) described
above to evaluate the microphysics, i.e., we obtain hydrometeor information from polarimetric radar observations and compare
it to the simulated hydrometeors. The focus is on high-impact weather events, i.e., hail and heavy rain. The goal is to exploit
the potential of polarimetric radar data for an evaluation of cloud microphysical schemes to statistically describe and discuss

90  the uncertainties of cloud microphysics in numerical weather prediction with respect to high-impact weather events.

The paper is structured as follows: Sect. 2 describes the measurement and simulation data. In Sect. 3, a model prediction
is presented using a convective case as an example to show that the model is fundamentally capable of producing realistic





predictions. Sections 4 and 5 then statistically evaluate the microphysical methods using heavy rain and hail/graupel events, respectively. Finally, Sect. 6 summarizes the results, draws conclusions, and discusses possible next steps.

## 2    Data and methodology

### 2.1    Evaluation periods

A total of 30 convective weather days in 2019 and 2020 were used for comparison. A detailed description of the radar observation and model simulation dataset can be found in Köcher et al. (2022). It is also summarized below.

The observational data is provided by the C-band radar in Isen, southern Germany near city of Munich, which is operated by the DWD. The radar is fully dual-polarimetric, therefore, polarimetric quantities such as horizontal reflectivity ($Z_{\mathrm{h}}$), differential reflectivity ($Z_{\mathrm{dr}}$), specific differential phase ($K_{\mathrm{dp}}$) and cross-correlation coefficient ($\rho_{\mathrm{hv}}$) are available. As part of the operational national radar network, the observation strategy is fixed: with a repetition rate of 5 minutes, a volume scan is performed, consisting of 11 PPI scans at elevation angles from 0.5° to 25° over the entire 360° azimuth. More information on the measurement strategy can be found in Helmert et al. (2014). This is the same data set used in Köcher et al. (2022, referred to as "Strategy A"). Further radar characteristics of the Isen radar and the exact days of measurement are listed in Table 1 and Table A1, therein. The Weather Research and Forecasting Model (WRF, Skamarock et al., 2019, version 4.2) employing 5 different microphysical schemes of different complexity (Thompson 2-mom: Thompson et al. (2008), Morrison 2-mom: Morrison et al. (2009), Thompson aerosol-aware: Thompson and Eidhammer (2014), spectral bin (SBM): Shpund et al. (2019), particle property prediction (P3): Morrison and Milbrandt (2015)) is used for the model simulations. The inner Munich domain has a grid spacing of 400 m and covers 144 km x 144 km. Only the inner third is used for analysis to exclude possible boundary issues. A radar forward operator (CR-SIM; Oue et al., 2020), consistent with the corresponding microphysical scheme, is applied to the model output to simulate the same polarimetric radar signals as observed: $Z_{\mathrm{h}}$, $Z_{\mathrm{dr}}$, $K_{\mathrm{dp}}$ and $\rho_{\mathrm{hv}}$ . Attenuation effects are simulated by the radar forward operator and applied to the simulated (differential) reflectivity to obtain attenuated (differential) reflectivities from the model output. Simulated and measured radar signals are then converted to a regular Cartesian grid with a grid spacing of 400 m using inverse range interpolation. This interpolation includes the four nearest data points, weighted by their distance $(1/\mathrm{distance})^2$ .

### 2.2    Hydrometeor classification

Polarimetric radar signals can provide information on the type of hydrometeors present in the measurement volume. There are hydrometeor identification (HID) algorithms that use polarimetric radar signals to determine the dominant hydrometeor species. In this study, the algorithm of Dolan et al. (2013) for C-band radar is applied for this purpose. This algorithm uses a fuzzy logic approach (Zadeh, 1965) based on theoretical scattering simulations using the T-matrix (Barber and Yeh, 1975) and Mueller-matrix (Vivekanandan et al., 1991) from Dolan and Rutledge (2009), originally for X-band. Based on the polarimetric radar variable ranges derived from the T-matrix scattering simulations, the method defines fuzzy logic membership functions





for each hydrometer type. These functions are then used to calculate a score describing how well the input radar signals match

a hydrometeor type. There are 10 different categories available, of which only specific classes are used in this study: drizzle and rain, hereafter referred to as "rain," and high density graupel, low density graupel, hail, and melting hail/large drops, hereafter referred to as "hail/graupel." Not used are: vertical ice, wet snow, aggregates and ice crystals. The hydrometeor classification is unable to distinguish between melting hail and large drops, and therefore uses a common class for both. We consider this class to be part of the hail/graupel category because the melting hail/large droplet classification typically always relates to hail

events and is typically associated with very large reflectivities. For classification, the HID algorithm uses four radar moments: reflectivity $Z_\mathrm{h}$, differential reflectivity $Z_\mathrm{dr}$, specific differential phase $K_\mathrm{dp}$, and correlation coefficient $\rho_\mathrm{hv}$. These variables are available from both the observed radar data set and our simulations after applying the CR-SIM radar forward operator. Accordingly, the HID algorithm is applied to both simulated and observed radar signals in the same way.

## 3 Demonstration of model output with an example case

Before systematic statistical examination of model capabilities, we will use an example to show that the models can generally provide realistic weather forecasts. Figures 1 and 2 show simulated and observed reflectivity, and the corresponding hydrometeor classification respectively, using one convective case in summer 2019. In convective situations this demonstrates that all simulations are capable of producing reasonably realistic weather forecasts, but at the same time it also shows the limitations of weather simulations in these situations. The case was chosen because convective cells were present over Munich at this time

step in all simulations and in the observations.

The weather situation chosen as an example occurred at 12:45 UTC on July 7, 2019, and was characterized by widely scattered convective showers over the Alps, directly over Munich, and in the Munich vicinity (Fig. 1). In general, all simulations at this time produce precipitation over much of the model domain. The general convective nature of the precipitation is correctly reproduced; there are numerous scattered convective cells, some isolated, in all simulations. The magnitude of the reflectivity

maxima is similar to the radar observations. However, this case also shows some limitations of numerical weather prediction for convective situations: the location of the simulated convective cells does of course not exactly match the observations. The area covered by precipitation is larger than that observed in most simulations. This is mainly due to a simulated precipitation area northeast Munich that was not observed at that time. Further, the simulated cells are smaller and more frequent, especially in the two Thompson simulations. This is not a general problem with the schemes: the total number of simulated convective

cells from the Thompson schemes over all 30 days is similar to those observed, as shown in Köcher et al. (2022). It rather points out the challenge to compare observations and simulations on a case study.

The corresponding hydrometeor classification for the same case is shown in Fig. 2. Most of the signals are classified as rain or drizzle in both the observations and all of the simulations. Embedded areas of large drops/melting hail or hail are classified in the observations and in four of the simulations: Thompson 2-mom, Thompson aerosol-aware, Morrison 2-mom, and P3, the

simulations are for the most part even able to produce hail cores of very similar size to those observed at this time. At the same time, general limitations of model predictions are also evident here: the exact location of the hail and the associated convective





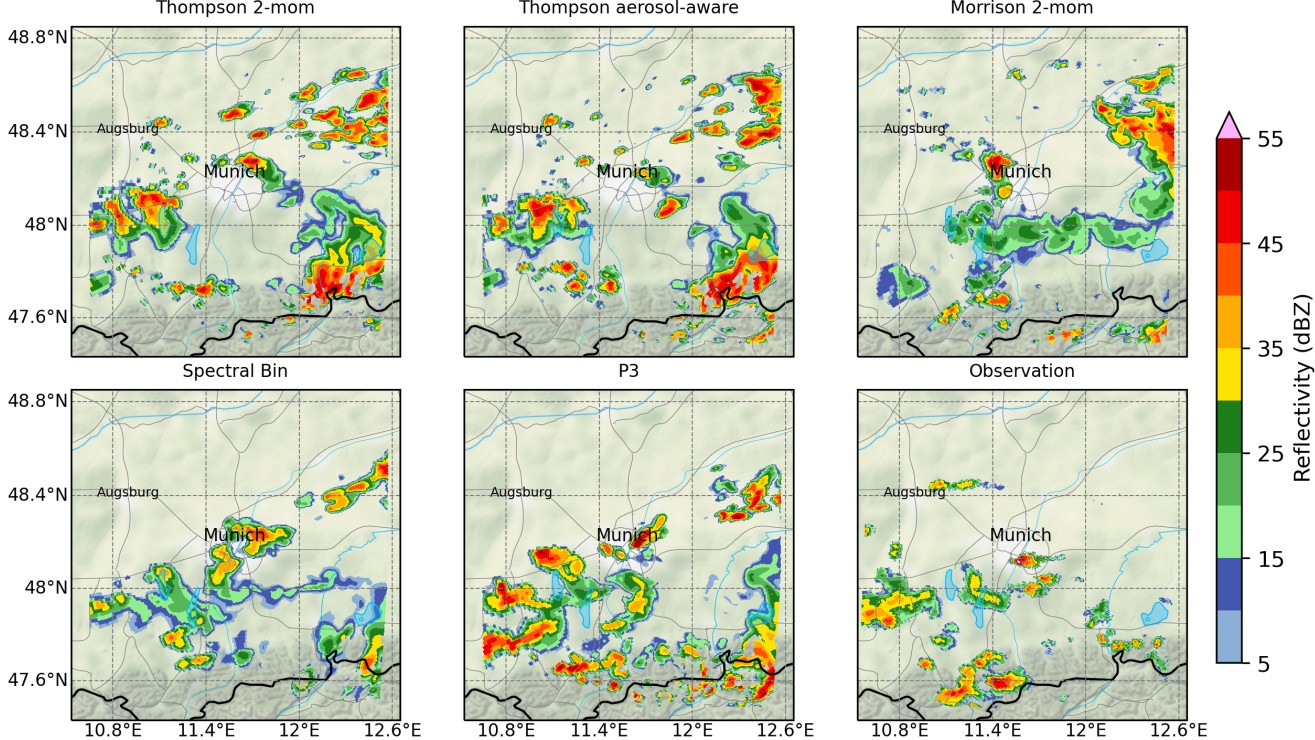

**Figure 1.** Simulated and measured reflectivity at July 7th, 2019, 12:45 UTC over the full domain size with a grid spacing of 400 m. Simulated reflectivity from the WRF model output after applying the CR-SIM forward simulator Oue et al. (2020). Background map tiles by Stamen Design (Stamen Design, 2022). Background map data by OpenStreetMap (OpenStreetMap, 2022, © OpenStreetMap contributors 2022. Distributed under the Open Data Commons Open Database License (ODbL) v1.0.). Roads, rivers, and lakes made with Natural Earth (Natural Earth, 2022).

cell is shifted compared to the observations and also varies depending on the scheme. Not all simulations classify (melting) hail at this time step. In addition, the hail core is partially classified as dry hail in the observations, while the simulations produce mostly melting hail at this time step. This scene shows that the choice of microphysics scheme has noticeable effects on the prediction of location, time, strength, and type of convective precipitation. But is this also relevant over a larger statistic? For a general evaluation of model prediction as a function of microphysics scheme, a statistical analysis over a longer time period is needed.

## 4 Heavy rainfall statistics

The objective of this study is to statistically evaluate microphysical models using polarimetric radar observations. To enable comparison between model and radar, either a radar signal must be simulated based on the model output (approach 1), or





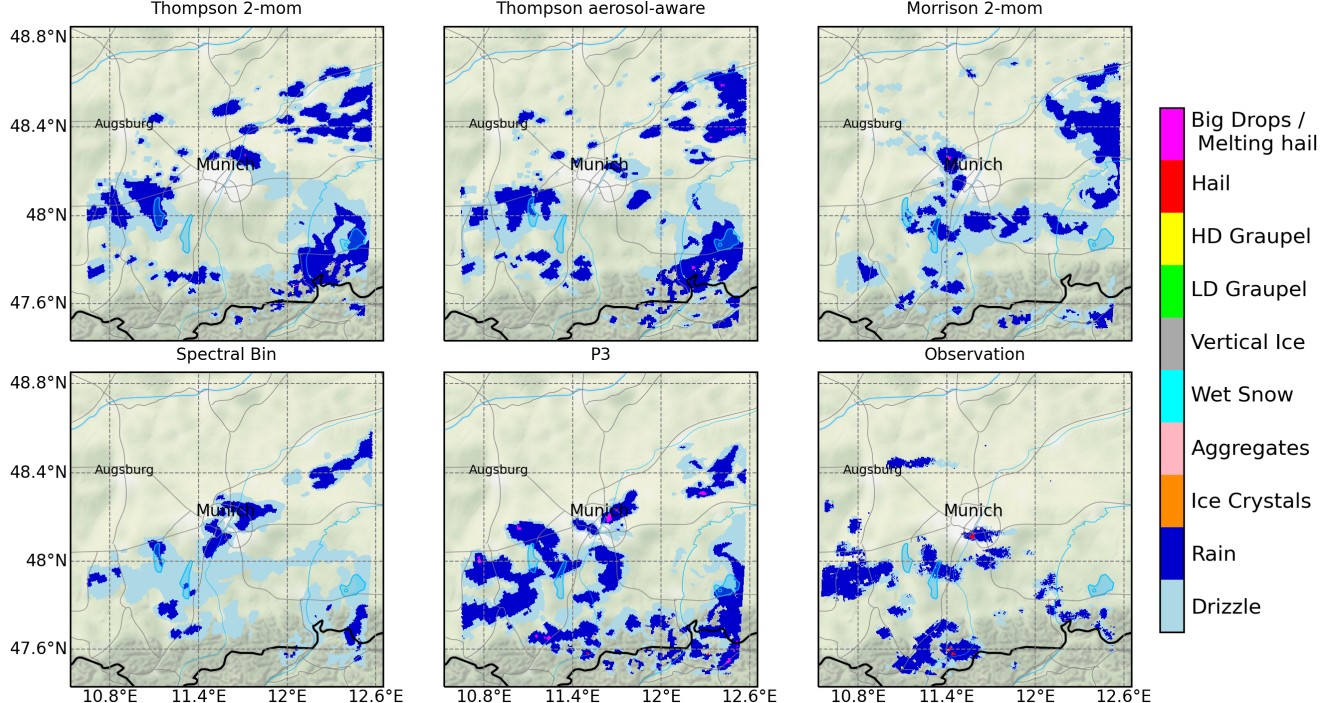

**Figure 2.** Same as Fig. 1, but for hydrometeor classification data, retrieved from the (simulated) polarimetric radar signals with Dolan et al. (2013). Background map tiles by Stamen Design (Stamen Design, 2022). Background map data by OpenStreetMap (OpenStreetMap, 2022, © OpenStreetMap contributors 2022. Distributed under the Open Data Commons Open Database License (ODbL) v1.0.). Roads, rivers, and lakes made with Natural Earth (Natural Earth, 2022).

information about hydrometeor classes must be obtained from the observed radar signals (approach 2). These two approaches are combined here to statistically compare observations and model outputs to evaluate microphysical processes related to high-impact weather events: heavy rain and hail. To define a heavy rain or hail event, we applied the hydrometeor classification of Dolan et al. (2013) (approach 2) to the observed and simulated (with a radar forward operator, approach 1) polarimetric signals

in the same way. The frequency and area of heavy rain and hail events defined in this way are then statistically compared to analyze differences between model and observation and the influence of the microphysical processes. The statistics extend over 30 convection cays. Since we do not have ground-based observations available due to radar elevation, we restrict all analyses to an altitude level of about 1 km above Munich. This altitude is the lowest possible altitude at which we have complete radar coverage of Munich.



## 4.1 Statistics based on observed and simulated reflectivity

We begin our statistical analysis with an evaluation of the frequency and area of heavy rain events. The top row in Fig. 3 shows the frequency and area or rain events for different event strengths at an altitude of 1 km above Munich. The strength of the event is defined by the simulated or observed reflectivity. We calculate the total duration of a day when rain was classified above a certain reflectivity threshold. For our 30 day dataset, this gives then a time series of 30 values. The minimum, mean, and maximum of this time series are shown on the top left of Fig. 3 for both the simulations for the five microphysical schemes and the observations. Thus, one can compare the frequency of rain of different intensities. The maximum possible time is 24 hours, which would correspond to rain for an entire day (for all 288 5-min steps) above the given reflectivity threshold. For this figure, it does not matter how large the area was classified as rain during a time step as long as at least one pixel was classified as rain. The missing information about the area of rain events is presented in the top right of Fig. 3. The term "area" here refers to the cross section area at 1 km altitude that was covered by the rain during a time step. The maximum possible area in the figure is $1800 \ \mathrm{km}^2$, which would mean that the Munich domain is completely covered by rain above the specified threshold. Both model output and radar observations provide data every five minutes. Figure 3 shows the minimum, mean, and maximum area over all time steps where rain was classified above the corresponding threshold. The total number of time with any rain somewhere in the domain averages to more than 13 hours in all models and in the observations. This high number of rain events is a consequence of the fact that our data set consists specifically of days with convective precipitation. At the smallest reflectivity thresholds (5 and 10 dBZ), all models underestimate the occurence of rain in the domain by about 5 hours on average per day compared to radar observations. This weak precipitation is typically classified as drizzle. At the same time, the area of these drizzle events is larger in the models than in the observations. This is due to the fact that the radar observations are much more likely to show scattered and isolated grid cells classified as drizzle, while the models typically show somewhat larger, contiguous fields of precipitation. In part, this difference may be due to some clutter that could not be filtered out from the observations. In any case, these observed isolated drizzle clouds are usually very small and likely evaporate before they reach the ground. There are some arguments why these drizzle events are nevertheless important, even if the precipitation does not reach the ground, such as by affecting the water balance and turbulent dynamics (Wyant et al., 2007). In addition, drizzle is often poorly represented in NWP models (Wyant et al., 2007; Wilkinson et al., 2012). However, we do not consider this issue to be the focus of this study and will not discuss it in detail.

The gray vertical lines in Fig. 3 show the thresholds used by the DWD for heavy rainfall ($15 \ \mathrm{l/m}^2$, $25 \ \mathrm{l/m}^2$, and $40 \ \mathrm{l/m}^2$; Deutscher Wetterdienst) after applying a Z-R relationship for convective precipitation (Woodley, 1970):

$$z = 300 \cdot R^{1.4}, \tag{1}$$



**Figure 3.** Frequency (left column) and area (right column) of classified rain above various thresholds. Minimum (dashed-dotted line), mean (solid line), and maximum (dashed line) over the 30-day data set. Gray vertical stripes: thresholds for heavy precipitation (15, 25, and 40 l/m$^2$) from the DWD after conversion to reflectivity (dBZ) using a Z-R relation. First row: statistics based on simulated and observed reflectivity thresholds at 1 km altitude. Simulated reflectivity from WRF model output after applying the CR-SIM forward simulator Oue et al. (2020). Second row: statistics based on mixing ratio thresholds at 1 km altitude. Third row: statistics based on thresholds for mixing ratio at the surface. Rain mixing ratio is direct WRF model output.





where $R$ is the rain rate in $1/\mathrm{m}^2$ defined by the DWD and $z$ is the reflectivity in $\mathrm{mm}^6/\mathrm{m}^3$, which is then converted into logarithmic units in decibels:

$$Z = 10 \cdot \log_{10}(z). \tag{2}$$

The heavy rain thresholds of 15, 25 and 40 $1/\mathrm{m}^2$ are thus converted to reflectivity thresholds of 41.2, 44.3 and 47 dBZ. Thus one gets an orientation at which reflectivities one is in the area of heavy rain. In these ranges the cloud microphysics has a strong influence on the simulated rain events. The frequencies from the simulated heavy rain events scatter around the observed frequency, which is pretty much in the middle of the different model simulations. However, the scatter is considerable, i.e., some of the simulations differ by 4 h in mean, which corresponds to a factor of more than 2 for the 41.2 dBZ - heavy rainfall threshold and a factor of about 5 for the 44 dBZ threshold. All simulations produce rainfall areas that are, on average, larger than the observed rainfall areas, almost regardless of the reflectivity threshold. Interestingly, the same microphysical schemes that simulate heavy rain events most frequently also simulate the largest rain areas, i.e., the simulations using the P3 scheme and the two Thompson schemes.

The most important information from this section is that most models produce heavy rainfall that is too large in area by a factor of up to four. The frequency of heavy rain events is more similar to observed events, but still scatters by a factor of up to 2 around the observations. In particular, for both Thompson schemes and the P3 scheme, this is a large overestimate of heavy precipitation events. These statistics are based on reflectivity thresholds. The question here is: will this also affect the amount of precipitation, i.e., rain mass? Reflectivity is disproportionately dominated by large droplets compared to rain mass. So in order to relate these results to actual rain mass, we repeat the analysis from this section, but based on thresholds for rain mass (or rain mass mixing ratio) instead of reflectivity.

### 4.2 Heavy rainfall statistics based on model mass mixing ratio

In the previous part we have shown statistics of rain events of different intensity based on HID fields classified from (simulated) radar reflectivity fields. To constrain the model output with radar observations it is necessary to simulate reflectivities from the model output. A disadvantage of this procedure is that it is relying on the radar forward operator. However, information about the simulated hydrometeors is also directly available in the model output in the form of the mixing ratio. In the following part, we repeat the analysis from the previous part with rain mass mixing ratio thresholds instead of reflectivity thresholds, to show how the analysis in reflectivity space is related to a direct analysis of the model output rain mass.

The middle row in Fig. 3 shows the same analysis as the top row, with the difference that this time the rain events are defined by the mixing ratio directly from the model output. The center left image in this figure shows how often rain was simulated above different mixing ratio thresholds. The center right image shows the covered area, following the same methodology as in the previous part. The analysis also takes place at an altitude of 1 km above the surface to guarantee comparability, even though we have the simulated mixing ratio data available at the surface as well. Most of the schemes produce distributions of rain events with a similar pattern over the different mixing ratio thresholds, actually only the Morrison scheme deviates noticeably and





produces rain events of any magnitude (with respect to the mixing ratio) more frequently than the other schemes. At higher mixing ratio thresholds ($> 10^4$ kg/kg), these events are also simulated over larger areas in the Morrison simulations than in the other simulations. This ultimately means that the Morrison simulations produce a significantly larger mixing ratio, and especially so for the particularly intense precipitation events.

When comparing this to top row, we find astounding differences: the schemes that overestimated heavy precipitation events the most when based on reflectivity thresholds (P3, Thompson 2-mom, Thompson aerosol-aware) simulated actually the least often heavy precipitation events when based on direct mixing ratio thresholds. Especially the conclusion that the P3, Thompson 2-mom and Thompson aerosol-aware simulations simulate too much heavy precipitation is not visible at all from the model mixing ratio directly. How can this discrepancy be explained? The following two reasons account for this: (1) Mass mixing

ratio depends to the third power ($\propto D^3$) on particle diameter, while reflectivity depends to the sixth power ($\propto D^6$) on particle diameter. Thus, particle size distribution plays an important role in inferring mass from reflectivity or vice versa. For example, particle size distributions with many small particles and few or no large particles may contribute significantly to the mass mixing ratio, but very little to reflectivity. (2) The HID algorithm always returns the dominant hydrometer class and not the mixture of all particles present in the volume. In contrast, the mixing ratio output by the model takes into account all simulated

hydrometeors. This means that if rain is common in a scheme but is rarely the dominant class, then rain is strongly represented in mixing ratio analyses but poorly represented in the HID analysis.

### 4.3   The role of the particle size distribution

To test the hypothesis from above, we calculated the rain particle size distributions (PSDs) for all our microphysical schemes. The mean particle size distributions over all time steps and all inner domain grid boxes at 1 km are shown in Fig. 4. The

spectral bin simulations provide total mixing ratios for a number of droplet size bins as direct model output. From this, the number concentration of raindrops of a given size bin is calculated by dividing the given mixing ratio by the mass of a droplet of the corresponding size bin. For the bulk schemes, we calculated the PSD according to the parameterization of the schemes by implementing the parameterization as described in the corresponding publication (Thompson et al., 2008; Morrison et al., 2009; Thompson and Eidhammer, 2014; Shpund et al., 2019; Morrison and Milbrandt, 2015) as well as directly from the code

available on github (Skamarock et al., 2019). The bulk schemes do not actually have fixed size bins, the number concentration can be calculated for any droplet size. To make the distributions comparable, we calculated the number concentrations for the bulk schemes for the droplet sizes that match the raindrop size bins of the SBM scheme. Figure 4 shows the PSD over these 18 bins. It can be seen that, on average, the SBM and Morrison schemes produce raindrops of 1 mm and smaller much more frequently (by a factor of more than 10) for small diameters. The opposite is true for large raindrops of 4 mm and larger.

These large raindrops are simulated not at all by the SBM scheme. We therefore believe that the PSD is the main reason for the different behavior of the schemes in reflectivity space and mixing ratio space. The sheer amount of small droplets in the Morrison and SBM schemes thus contributes noticeably to the total mass, but less to the reflectivity, which is simulated much lower in comparison especially in the SBM scheme due to the missing large raindrops.





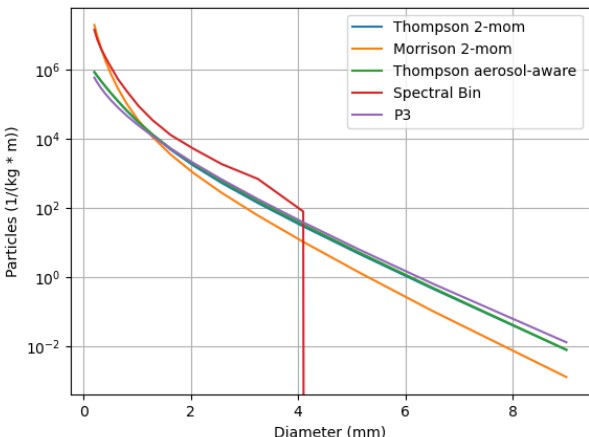

**Figure 4.** Mean rain drop size distribution over all 30 days at 1 km altitude.

This strongly suggests that it is not the rain mass produced that is the problem with the simulations, but rather the distribution

of mass across droplet sizes. We cannot say with certainty what the particle size distributions looked like in reality during our
30-day data set because we do not have direct measurements of the droplet size distribution. But given that the SBM scheme
simulate high mixing ratio of rain mass, but at the same time produce too few heavy rain events based on the reflectivity
produced, it stands to reason that this scheme generally produce too few large raindrops. The exact opposite is true for the P3,
Thompson 2-mom, and Thompson aerosol-aware simulations. These results are consistent with the findings of Köcher et al.

(2022), where the simulated differential reflectivity was statistically compared with the observed one. In particular, the SBM
scheme did not produce larger differential reflectivity in the lower elevations, suggesting the absence of large drops in the SBM,
while the P3 and Thompson schemes produced large differential reflectivity signals too frequently. A similar comparison of
polarimetric radar signatures is performed by Wu et al. (2021) for a typhoon precipitation event in 2016. They note that none of
their simulations were able to successfully reproduce the observed polarimetric radar signatures, and attribute this to too large

median raindrop sizes in the Morrison and Thompson schemes, while at the same time the frequency of very large raindrops
in the Thompson scheme is lower than observed. In contrast, Putnam et al. (2016) find that both Morrison and Thompson
2-mom produce reflectivity values that are too high, which they attribute to PSDs containing too many large drops, too much
precipitation coverage, and, in the case of the Morrison simulations, a bias due to wet graupel. With respect to our study, we
can confirm too much precipitation coverage, and our results suggest that there are too many large raindrops in the Thompson

simulations, which is consistent with Putnam et al. (2016) but in contrast to Wu et al. (2021). However, both studies evaluate the
microphysical schemes using only case studies, which is not generally applicable to different weather situations. This shows
an advantage of our statistical approach, which allows more robust conclusions.

The analysis to this point has been limited to an altitude of 1 km above the surface, limited by the radar observations.
However, another question remains: Does the analysis at 1 km altitude translate to precipitation at the surface? Precipitation



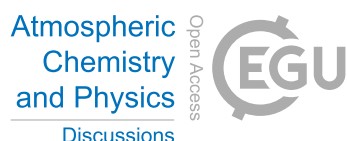

on its way to the ground is affected by processes such as evaporation, droplet fall velocity, and droplet size distribution, and different microphysical methods treat these processes differently. Therefore, we repeat the mixing ratio analysis again, but with data from the surface, to relate the results from 1 km altitude to results at the surface.

### 4.4 Rain mass mixing ratio analysis at the surface

The bottom row in Fig. 3 shows the same analysis as the center row, except that this time surface mixing ratios are analyzed instead of 1 km altitude. We find that at the surface, the Morrison and SBM simulations showed the most frequent and most widespread rain events, throughout most mixing ratio thresholds. This is in general agreement with the analysis at 1 km height. However, the difference, especially between Morrison and the other schemes is getting smaller. This suggests that rain within the Morrison schemes is undergoing stronger evaporation compared to the others. Given that the Morrison scheme produced on average the highest number of very small rain drops of 0.5 mm and smaller at 1 km altitude, a high evaporation is to be expected. Smaller droplets evaporate easier, due to high surface tension. Since the general ranking between the schemes is almost the same between 1 km altitude and the surface, we argue that findings in 1 km altitude are a good proxy also for the surface. However, the difference between the schemes at the surface definitely becomes smaller, all schemes produce frequency and area of rain events of similar magnitude.

Summarizing the results of the rainfall statistics, we can note the following: all schemes overestimate the area of heavy rain events based on reflectivity thresholds, P3 and the two Thompson schemes also overestimate the frequency of heavy rain events, while the SBM and Morrison schemes underestimate the frequency of heavy rain events. Further analysis of the rain mixing ratio and the calculated particle size distributions indicate that large raindrops that contribute strongly to high reflectivities are simulated too frequently in the P3, Thompson 2-mom, and Thompson aerosol-aware simulations, while large raindrops occur too infrequently in the SBM simulations.

## 5 Hail and graupel statistics

So far, the focus has been exclusively on heavy precipitation events. However, hail events are also weather events with damage potential and therefore of interest. Therefore, in this section we focus on hail statistics based on the same data set.

### 5.1 Statistics based on observed and simulated reflectivity

The applied HID algorithm from Dolan et al. (2013) distinguishes between hail and graupel in the classification. In contrast, the microphysical cloud schemes in WRF applied in this study do not distinguish between hail and graupel and only predict mixing ratios and number concentrations of graupel. However, it is quite possible that hail is classified from the simulated radar signals, presumably primarily as a result of large graupel particles, as large graupel and hail produce similar radar signals. Therefore, we combine the classified hail and graupel particles into a common class. Figure 5 shows the area and frequency of these hail/graupel events in the same manner as Fig. 3 for rain. The choice of the microphysics schemes has a significant impact on these hail/graupel statistics, across all reflectivity thresholds. The most extreme case is the SBM scheme, which hardly produces





any hail/graupel. There is not a single time step within the 30 day dataset at which the SBM scheme simulated hail/graupel grid cells of 35 dBZ or higher (Top left image in Fig.´ 5). This is due to the fact that the simulated reflectivity for the SBM scheme is generally much lower compared to the other schemes. However, a high reflectivity is required for the HID algorithm to classify graupel and hail. The same is true for the Morrison scheme, although not to the same extent. However, most of the

other schemes also consistently show fewer hail/graupel events compared to the observations. Unlike the rain analysis, this is consistent across all reflectivity thresholds and not limited to the lower reflectivities. Only the P3 scheme is similar in terms of frequency and for it only the highest reflectivity events are too rare. Since none of the simulations, regardless of the cloud microphyics scheme, were able to reproduce these extreme events, we do not believe that this is related to the microphysics scheme, but rather a consequence of the model grid resolution. Such high reflectivities require very large particles. For hail

formation, for example, strong updrafts must be present. There is some discussion about the grid spacing required to properly represent these updrafts. Lebo and Morrison (2015)) and Jeevanjee (2017), for example, show that a grid spacing of at least 250 m is required before some convective storm characteristics, such as vertical velocity or convective core areaconverge, i.e., further decreasing the grid spacing has only a limited effect. This means that even with our grid spacing of 400 m, which is much better than current weather models, may still be slightly too small to correctly simulate the strongest hail events.

We note, however, that most models are quite capable of producing events of this magnitude at reflection thresholds of about 45 dBZ and 50 dBZ. However, most models underestimate the frequency and area of these events compared to radar observations. With a reflectivity threshold of 50 dBZ (which may be, in part, hail events), the P3 scheme is the only one capable of producing similar statistics in terms of area and frequency of hail/graupel events. The main difference of the P3 scheme from the other schemes is that instead of predicting the moments of multiple ice hydrometeor classes, the P3 scheme uses only one

ice class and instead predicts the properties of that ice class, such as the fraction of rime mass. This is more flexible and may better reflect the variability of real ice particles. While the Thompson 2-mom and Thompson aerosol simulations were at least able to reproduce the hail/graupel statistics at lower reflectivity thresholds, Morrison and SBM produce too few and too small hail/graupel events regardless of reflectivity threshold. The reason for this is likely differences in particle size distribution in both cases. Morrison and SBM generally produce much lower reflectivity values, presumably because again the larger particles

are absent.

Again, however, the question arises as to how statistics based on radar reflectivity translate to statistics based on mass mixing ratio. Therefore, we continue with the analysis of hail/graupel in terms of mixing ratio, following the same structure as in the section on rain.

## 5.2 Hail and graupel statistics based on model mass mixing ratio

The middle row of Fig. 5 shows the same analysis as the top row, this time for the model's initial graupel mixing ratios. A clear ranking can be seen between the microphysical schemes, both for area and for the frequency of graupel events, which is almost independent of the mixing ratio threshold: the spectral bin simulations produced the most frequent and widespread graupel events on average, while the Morrison and especially the P3 schemes produced graupel events that were the least frequent and also the smallest. P3 is a special case here, because unlike all other schemes, the P3 scheme does not provide a graupel



**Figure 5.** Same as Fig. 3, but for hail and graupel statistics. Frequency (left column) and area (right column) of hail/graupel events above various thresholds. Minimum (dashed-dotted line), mean (solid line), and maximum (dashed line) over the 30-day data set. First row: statistics based on simulated and observed reflectivity thresholds at 1 km altitude. Simulated reflectivity from WRF model output after applying the CR-SIM forward simulator Oue et al. (2020). Second row: statistics based on mixing ratio thresholds at 1 km altitude. Third row: statistics based on thresholds for mixing ratio at the surface. Graupel mixing ratio is direct WRF model output. In case of the P3 scheme, rime ice mass mixing ratio is used instead of graupel mass mixing ratio.





mixing ratio, but a rime ice mass mixing ratio, which we use instead. This means that also only partially rimed ice (and thus by definition not graupel) is included in the evaluation. Therefore, the P3 lines in the center and bottom row in Fig. 5 should be considered as the maximum and not the actual graupel events. Interestingly, regardless of this difference, the P3 line is lower than all the other lines, which means that less frequent and smaller events are simulated in P3 compared to the other schemes, even when any rimed particle is considered. Comparing the mixing ratio analysis to the frequency and area statistics based on

reflectivity in the previous section, we again see a stark contrast, especially for the P3 scheme, where graupel/hail events are most common when obtained from radar reflectivity, and for the SBM scheme, where graupel/hail events are least common. Only for the Morrison scheme, we find graupel/hail events less frequently, regardless of mixing ratio or reflectivity analysis. The SBM scheme, on the other hand, is again likely missing the larger particles, since a large mass of graupel particles is generated, but this does not translate into high reflectivities.

The graupel statistics from this section were performed at 1 km altitude to ensure comparability with radar observations. Again, the question is whether the results from this section can be extrapolated to the surface. Therefore, we repeat the mixing ratio analysis at the surface.

### 5.3 Hail and graupel mass mixing ratio analysis at the surface

The bottom row in Fig. 5 shows the same analysis as the middle row, except that this time the mixing ratio is analyzed at

the surface rather than at 1 km altitude. It can be seen that the SBM scheme at the surface simulates the most frequent and widespread graupel events, regardless of the reflectivity threshold, followed by the two Thompson schemes. The least frequent and also the smallest graupel events are simulated by the P3 scheme. This ranking is the same as at 1 km altitude, indicating that the findings from 1 km altitude are approximately applicable to the surface as well. However, the P3 scheme simulates much smaller areas of graupel at the surface than at 1 km altitude. This indicates that the melting process in the P3 scheme is is

stronger than in the other schemes. However, since we do not have measurements of the particle size distribution at the surface, we cannot say whether these high melting rates are realistic.

In summary, for the hail and graupel statistics, no model is able to reproduce the most extreme reflectivity statistics of greater than 55 dBZ, which is likely a resolution problem. Hail/graupel at reflectivity thresholds of 45 to 50 dBZ is correctly reproduced only by the P3 scheme. The other schemes, especially the Morrison and SBM, underestimate the frequency and

area of hail/graupel events regardless of reflectivity threshold. Analysis of the graupel mixing ratio directly from the model output suggests that the SBM scheme produces a high graupel mixing ratio but this is not correctly distributed and lacks the larger graupel particles.

### 6 Summary and conclusions

In this study, microphysical schemes of varying complexity are evaluated via statistical comparison with polarimetric C-

band radar observations. The focus is on the statistics of high-impact weather events, i.e., hail and heavy precipitation. The dataset consists of 30 convective days during the summers of 2019 and 2020. The radar observations consist of polarimetric





volume scans from the C-band radar at Isen, which is operated by the German Meteorological Service (DWD). The same days are simulated using a convectivion-permitting Weather Research and Forecasting (WRF) Model setup over Munich with a horizontal grid spacing of 400 m, 40 vertical levels and 5 different microphysical schemes of varying complexity. The radar

forward operator CR-SIM (Oue et al., 2020) is applied to the model results and yields simulated polarimetric radar signals, consistent with the corresponding microphysical schemes. The hydrometeor classification algorithm of Dolan et al. (2013) is applied to simulated as well as observed radar signals to identify dominant hydrometeor classes and define heavy rain or hail events. Frequencies and areas of these events from the radar observations are then compared to the simulations to evaluate the microphysics schemes.

Analysis of the heavy rain events shows that all simulations, regardless of the microphysical scheme, overestimate the area of the heavy rain events based on the reflectivity thresholds compared to radar observations. Since this is independent of the microphysical scheme, we suspect that this is not an issue of the microphysics but rather related to the limited grid resolution: The model tends to produce larger, contiguous precipitation fields, whereas in reality precipitation fields are sometimes more scattered and small. With respect to the frequency of heavy rainfall events, there is significant scatter between simulations: The

P3, Thompson aerosol-aware, and Thompson 2-mom schemes overestimate the frequency of heavy rain events by a factor of up to 2 compared to radar observations, while the spectral bin (SBM) scheme underestimates the frequency of heavy rain events by a factor of up to 2. This means that the P3 and both Thompson schemes greatly overestimate both frequency and area of heavy rain based on reflectivity thresholds. To apply these results to rain mass, an analysis of the same statistics was performed based on the mixing ratio of rain mass in the model. In contrast to the reflectivity analysis, the P3, Thompson aerosol-aware,

and Thompson 2-mom models produce the fewest heavy rainfall events and the smallest in area. Analysis of the simulated rain particle size distributions shows that the Morrison and SBM schemes on average produce more small rain drops of 1 mm and smaller by a factor of up to 10, while at the same time the SBM scheme simulates the fewest large raindrops. We conclude that it is not the rain mass produced that is the problem, but rather the distribution across droplet sizes: compared to the radar-observed heavy rain events, the P3, Thompson 2-mom, and Thompson Aerosol-aware schemes produce large raindrops too frequently,

while the SBM simulations produce too few. The results related to the Thompson 2-mom and Morrison schemes are in conflict with a previous study by Wu et al. (2021), but are consistent with Köcher et al. (2022) and Putnam et al. (2016), highlighting the problem of evaluating microphysical schemes using case studies and demonstrating the importance of statistical evaluation as in this study.

Similarly, we repeated the analysis for hail and graupel events. We note that none of our simulations is able to reproduce the

radar observations at the highest reflectivity thresholds 55 dBZ and above. This is likely related to grid spacing issues. Our grid spacing is 400 m, which is much better than the typical resolution used for operational numerical weather predictions. However, realistic updrafts are required to produce hail large enough to produce reflectivities greater than 55 dBZ. Some studies suggest that grid spacing of 250 m or less is required for realistic convective updrafts (Lebo and Morrison, 2015; Jeevanjee, 2017). Therefore, we believe that our grid spacing of 400 m is still insufficient to correctly resolve the most extreme hail events.

However, at slightly smaller reflection thresholds of 45 dBZ to 50 dBZ, the models are partially able to simulate events of this magnitude. However, only the P3 scheme is able to reproduce the frequency and area of hail/graupel events of this magnitude





very closely to the statistics observed by radar. All other schemes, particularly the Morrison and SBM schemes, underestimate the frequency and area of hail/graupel events, regardless of reflectivity threshold. However, analysis of the model's graupel mixing ratio shows that of the SBM and Morrison schemes, only the Morrison scheme simulates a low mass of graupel on average. The SBM scheme, on the other hand, simulates the largest mass of graupel. Thus, we conclude that the SBM scheme has issues in distributing the mass correctly over the particle sizes; again, the large particles are missing.

Relating radar-derived weather statistics to precipitation statistics at the surface is challenging. Although the simulations suggest that the mixing ratio at 1 km altitude is strongly related to the mixing ratio - and hence precipitation rate - at the ground, there are processes such as evaporation, drop breakup, or self-collection that affect this. In special cases, it is certainly possible for these processes to significantly change the precipitation rate. Therefore, one always has to rely on a model that correctly simulates these processes when radar measurements are used to make statements about precipitation on the ground. Furthermore, we used a radar forward simulator for the analysis in this study. This is a useful tool to simulate the expected radar signals using the model data. However, it makes broad assumptions about the aspect ratio, orientation, shape, and density of the particles. Therefore, the application of these rigid relations inevitably results in the simulated particle properties not exhibiting the same variability as in nature. Furthermore, in this study, we applied a hydrometeor classification algorithm to classify the predominant hydrometeors in the radar observation volumes. This algorithm is based on theoretical scattering simulations and only the dominant hydrometeor class is determined. This makes it difficult to relate the radar-based results to precipitation rates. It would be very helpful if, in addition to the dominant hydrometer class, the corresponding mixing ratio was also derived from the polarimetric radar variables. Within a sub-project of the DFG (German Research Foundation) Priority Programme 2115 (PROM, Trömel et al., 2021), an HMC algorithm is currently being developed for this purpose, based on a clustering approach and an algorithm for quantifying the mixing ratio following Grazioli et al. (2015), Besic et al. (2016) and Besic et al. (2018), and aiming to calculate the mixing ratios of hydrometeor classes as well. Thus, mixing ratios derived from the radar signal could be directly compared with the mixing ratios simulated by the model.

Can numerical weather prediction (NWP) forecasts be further improved by applying more complex microphysics schemes? Our analysis of high impact weather events does not show a clear advantage of more complex schemes; there are significant deviations from observations for all of the schemes applied. Simply increasing the complexity does not necessarily improve predictions. Novel observations are needed provide the necessary information and better constrain the microphysical processes. In this work we have shown the potential of polarimetric radar observations to do just that and suggest they be further explored.

*Code and data availability.* The polarimetric radar data from the operational C-Band radar in Isen is available for research from the German Meteorological Service (DWD) upon request. Data of WRF and CR-SIM simulations are available from the authors upon request. The software developed for this paper is available at https://doi.org/10.5281/zenodo.7428844 (Köcher, 2022). The Weather Research and Forecasting model (WRF, version 4.2) is publicly available on GitHub at https://github.com/wrf-model/WRF (last access: 20 June 2020; https://doi.10.5065/1dfh-6p97, Skamarock et al. (2019)). The forward operator CR-SIM (version 3.33) is available on the website of Stony Brook University (https://you.stonybrook.edu/radar/research/radar-simulators/, Oue et al. (2020)). The hydrometeor classification is publicly



*Author contributions.*  GK developed the methodology presented and wrote the manuscript in its current form. TZ and CK supervised, discussed the scientific content and commented on the manuscript.

*Competing interests.*  The authors declare that they have no conflict of interest.

*Acknowledgements.*  We gratefully acknowledge the project IcePolCKa ("Investigation of the initiation of convection and the evolution of precipitation using simulations and polarimetric radar observations at C- and Ka-band") funded by the Deutsche Forschungsgemeinschaft (DFG, German Research Foundation) – 408027579 – as part of the special priority program on the Fusion of Radar Polarimetry and Atmospheric Modelling (DFG SPP-2115, PROM). We want to thank Stefan Kneifel for his comments on the manuscript.



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
