# Peer review of "Influence of cloud microphysics schemes on weather model predictions of heavy precipitation"

_Atmospheric Chemistry and Physics, 2022_

## Referee Comment (RC2)

**General comments.** This paper uses polarimetric radar observations to test simulations using various microphysics schemes for summer convective cases over Germany. A major strength of the paper is the statistical evaluation over many cases in contrast to the approach often employed of focusing on a single case study. This topic is important as the representation of microphysics is a major source of uncertainty in NWP models. The paper is reasonably well written and the conclusions seem to be robust. My main concerns on the science center on 1) consistency between the radar forward simulator and internal assumptions and characteristics within the microphysics schemes, and 2) inconsistencies in the analysis of P3 (primarily for graupel/hail). These concerns are detailed in major comments below. I also include several minor comments, mainly to clarify certain issues or improve the presentation. Finally, a handful of technical/editorial comments are included at the end. Overall, I think the paper could be acceptable if the main comments are addressed.

**Recommendation:** *Major revision*

**Major comments.**

1. One of my main comments concerns the consistency of assumptions and ice properties internal to the microphysics schemes with assumptions in the radar forward operator calculations using CR-SIM. It's mentioned a few times that these calculations are consistent, but in my view more detail is needed on this. For instance, what ice particle properties are needed or assumed by the forward operator calculations? I guess this includes particle density, size distribution information, and particle habit? Are there other inputs or assumptions made about ice particles in CR-SIM? I think it should be straightforward to couple the traditional bulk schemes (Thompson, Morrison) with CR-SIM. However, this seems less straightforward for coupling with SBM and particularly P3. In P3, the particle density (e.g., particle mass-size relation) varies across different regions of the size distribution. How was this accounted for? Particularly relevant for this paper, in all the schemes what is assumed for the distribution of liquid water on melting ice particles? This is particularly important in this paper given the focus on radar signatures of hail and particularly the reflectivity bias in all schemes in conditions of large hail (i.e., high reflectivities).

Perhaps these issues are discussed in some of the previous papers on CR-SIM, but more discussion is needed in this paper.

2. Figure 5 and ~p. 16. I think the analysis here is a misinterpretation of graupel and rimed ice as simulated by P3. Graupel-like or even hail-like ice particles in P3 don't necessarily consist wholly of rimed ice mass, -- they may contain substantial mass grown by vapor deposition as well (but still have high particle densities expected for graupel/hail). Thus, I think it's incorrect to only include rimed mass for the mixing ratio threshold in Fig. 5 for P3. If the hydrometeor ID identifies a particular time/location as hail/graupel from the P3 output, I would include all ice (rime plus vapor grown) in the analysis. This is particularly relevant to the analysis of graupel/hail since there is no separate rimed ice category in P3. It seems quite likely that ice

present at the 1 km level in P3 for summer convective cases will be fast falling and with a high rime fraction (and therefore hail-like) anyway.

3. I didn't see any discussion on the scale of the radar observations versus the model, other than discussion of 400 m grid spacing possibly explaining the low bias for the instances of high reflectivity associated with graupel/hail. Is there a general scale mismatch between the radar and the observations? Any scale mismatch is likely to be quite important for instances of heavy rain and hail. If the radar data is higher resolution, can it be appropriately averaged to give a comparable scale to the model data? Keep in mind that the effective resolution in models like WRF is about 5-7 times the grid spacing (Skamarock 2004, MWR).

4. The SBM results from Figure 4 look rather strange, in particular, the sharp drop at diameters just above 4 mm. What is the explanation for this? Does it have something to do with how breakup is treated in this scheme? Or is this the maximum size of bins in the scheme?

**Minor comments.**

Abstract, line 7. "heavy rain events" is imprecise. Can you state here the specific rain rates used to define such events?

Line 26. I disagree that increasing resolution eliminates problems caused by inaccurate parameterizations, it addresses *some* of these problems. Thus, I suggest adding "some" before "inaccurate parameterizations".

Line 32. Add "usually" before "predict", since not all bulk schemes assume a predefined statistical function for the particle size or mass distributions (a few schemes predict processes directly from moments without assuming any functional form of the size distribution, e.g. Kogan and Belochitski 2012, JAS).

Line 41. Not sure I agree that it is not known exactly which processes are poorly represented and I don't think the paper cited here (Morrison et al. 2020) makes this argument either – we have some idea of which processes are most uncertain or most poorly represented. Perhaps reword this sentence to "It is known that many processes, especially those involving ice microphysics, are poorly represented in numerical weather prediction models (Morrison et al., 2020)."

Line 151. I'd add "single" before "case study", as I think this states the point here better.

Line 184. Not clear what you mean by "The missing information about the area of rain events is presented in the top right of Fig. 3." What is missing? I think what you mean is simply "The area of rain events is presented in the top right of Fig. 3."?

Line 188. Confusing as written. Suggest removing "number of".

Line 240. Are the differences really "astounding"? Maybe "substantial" ort "major" would be better?

Lines 244-246. You might note that these relations of mass and reflectivity to diameter are true for liquid drops (or more generally, isometric particles).

Line 247. Technically, all of the bulk schemes here use complete size distributions, meaning they extend mathematically from 0 to infinity. Thus, it's better to just say "..few large particles may contribute significantly…" rather than "few or no large particles…".

Line 268. I think this could be reworded more clearly – I suggest replacing "due to the missing large raindrops" with "due to the lack of large raindrops".

Line 290. Technically droplet fall velocity and droplet size distribution are not processes. Suggest rewording this to "is affected by processes such as evaporation and drop sedimentation, …"

Line 300. They don't evaporate faster because of high surface tension, it's because evaporation in the schemes depends mainly on the number concentration times the mean radius (referred to as the integral radius), with some additional modification to account for ventilation.

Lines 315-316. This is not correct – the P3 scheme simulates not just number and mass mixing ratios of ice. By predicting additional ice attributes, it can distinguish between graupel-like and hail-like ice (for example, by differences in mean density, size, and fallspeed).

Lines 331-332. This is confusing. I'd reword to "a grid spacing less than about 250 m is required…".

Line 333. I'd add "when" before "further".

Line 339. Note that a multiple "free" category version of P3 exists (see Milbrandt and Morrison 2016, JAS). Thus, I'd reword this to "…this version of the P3 scheme uses only one ice class…".

Line 378. "likely" seems too strong of a word to use here. Perhaps reword to "which might be a resolution problem". Same comment on line 415.

**Technical/editorial comments.**

Line 4, abstract. Suggest adding "the" before "observation dataset".

Line 44. Perhaps replace "just this" with "such".

Line 78. Replace "are" with "were".

Line 84. Add "schemes" after "microphysics"?

Line 99. The first "is" should be "are" (data here is plural).

Line 179. I feel "then" could be removed.

Line 226. Replace "it is relying on" with "it relies on".

Line 227. "mixing ratio" should be "mixing ratios".

Line 260. I think "github" should be "GitHub"?

Line 260. Reword to "Because the bulk schemes do not actually have fixed size bins…"

Line 272. "simulate" should be "simulates". Alsop "produce" should be "produces". Same comment on the next line (line 273) as well.

Line 278. "is" should be "was" and "note" should be "noted".

Line 279. "attribute" should be "attributed".

Lines 277-281. This sentence is very long, perhaps break it up into 2 sentences.

Lines 281-286. I'd suggest using past tense in the writing here, since you're describing what previous studies found.

Line 297. Add a comma after "schemes". Also, replace "is getting smaller" with "becomes smaller".

Line 298. "schemes" should be "scheme".

Line 301. I think you can remove "also".

Line 312. Add "are" before "of interest".

Line 331. There's an extra right parenthesis after "(2015)".

Line 332. Space is missing between "area" and "converge".

Line 333. Remove "with".

Line 335. I think "reflection" should be "reflectivity"?

Line 338. I'd replace "from" with "with".

Line 343. Replace "distribution" with "distributions".

Line 374. Typo: "is" is repeated twice.

Figure 3 caption. "Oue et al. (2020)" should be "(Oue et al., 2020)". Same comment with the Figure 5 caption as well.

**References.**

Kogan, Y. L., and A. Belochitski, 2012: Parameterization of cloud microphysics based on full integral moments. *J. Atmos. Sci.*, 69, 2229-2242.

Milbrandt, J. A., and H. Morrison, 2016: Parameterization of cloud microphysics based on the prediction of ice particle properties. Part 3: Introduction of multiple free categories. *J. Atmos. Sci.*, 73, 975-995.

Skamarock, W. C., 2004: Evaluating mesoscale NWP models using kinetic energy spectra. *Mon. Wea. Rev.*, 132, 3019-3032.

---

## Author Comment (AC1)

This is an interesting study giving a statistical evaluation of different microphysical schemes using real radar observations on 30 convective cases.

The first part of the results dedicated to the analysis of heavy rain is very convincing. The two ways of sorting the results as a function of reflectivity first and of rain content next is nice and helps showing the effect of the understimation or overestimation (depending on the scheme) of the number of large drops in the PSD.

Regarding the part with hail and graupel statistics, I don't always agree with the interpretation of the results.

Thank you for this review. Our point-by-point review is in blue. The changes made to the manuscript are highlighted in red. A marked up-version showing all changes to the manuscript is provided along with the revised manuscript.

l 317: "large graupel and hail produce similar radar signals"

It is true only if graupel and hail are modeled with the same characteristics (PSD and density for Zh). Depending of the density options for graupel in the microphysics scheme you are evaluating, you could underestimate the maximum possible reflectivities that would be reached if explicit with hail (which is denser than graupel) was modeled in your schemes.

There are several instances where the classification algorithm identifies hail even though the original model does not simulate it. We still believe that these "false" identifications are primarily due to large graupel particles, since they are similar in shape to hail and therefore have similar differential reflectivity, and large graupel particles can also produce large reflectivities. However, it is true that hail would produce even higher reflectivities due to its higher density. We did, in fact, change the 'graupel/hail' analysis to a general 'ice' analysis, to allow a fairer comparison between the P3 scheme and the other schemes. Therefore, this sentence is not required anymore and we deleted it completely.

l 327: "Since none of the simulations, regardless of the cloud microphyics scheme, were able to reproduce these extreme events, we do not believe that this is related to the microphysics scheme, but rather a consequence of the model grid resolution."

The resolution very probably plays a role but again, you can't simulate the extreme reflectivities due to hail (in your observation) while you don't explicitly have hail in your model (and the corresponding options in the forward operator).

It is true that we underestimate the maximum possible reflectivity if we do not explicitly calculate hail (which has a higher density than graupel). This has not been taken into account so far. We have added a part that takes into account the influence of density on the simulated reflectivity. The exact wording can be found in our response to the next comment.

L 363: "The SBM scheme, on the other hand, is again likely missing the larger particles, since a large mass of graupel particles is generated, but this does not translate into high reflectivities"

Again, could this be also due to a different graupel density in this scheme compared to others ?

More information about the differences in the density of graupel / rimed fraction between the different schemes (and compared to typical hail density) should be included in the paper, either to evaluate if this could have an effect or not.

The difference in graupel density between the SBM scheme and most of the other schemes is not drastically different. The Morrison and SBM scheme use 400 kg m$^{-3}$ and the two Thompson schemes use 500 kg m$^{-3}$ as a constant graupel density. The P3 scheme uses a varying ice density, that can reach up to 900 kg m$^{-3}$. We don't think that the graupel density of the SBM scheme is the main reason for the low reflectivity produced, given that the density is not drastically different to most other bulk schemes. However, we understand that the density is playing a major role in general when simulating reflectivity, and should be discussed more in detail. Especially for the P3 scheme, it was not yet acknowledged that the more flexible density assumptions (reaching up to 900 kg m$^{-3}$, i.e., hail-like particles) might produce more realistic reflectivities compared to the observations.

We acknowledge the density influence now at multiple instances:

Section 5.1:
> *(2) Particle density: The particle density strongly influences the reflectivity. All schemes except the P3 scheme consider graupel particles with constant density of 400 - 500 kg m$^{-3}$, and do not explicitly calculate hail. Hail particles, however, are typically much denser than graupel. This means, if hail events are observed, the high hail density can lead to high observed reflectivities that cannot be reproduced by the models due to the lower assumed particle density. Only the P3 scheme has a more flexible approach that allows varying ice particle density reaching up to 900 kg m$^{-3}$.*

Section 5.1:
> *The P3 is also the only scheme that allows ice particles to reach densities up to 900 kg m$^{-3}$, i.e., to simulate hail-like particles.*

Section 5.1:
> *However, graupel density assumptions could also play a role. Both, Morrison and SBM assume a graupel density of 400 kg m$^{-3}$ which is slightly lower than assumed by the Thompson schemes with 500 kg m$^{-3}$.*

Section 6:

*This might be related 1) to a resolution problem, 2) to density assumptions that are not representative for high density hail-like particles or 3) the absence of partially melted particles in the simulations.*

---

## Author Comment (AC2)

**General comments.** This paper uses polarimetric radar observations to test simulations using various microphysics schemes for summer convective cases over Germany. A major strength of the paper is the statistical evaluation over many cases in contrast to the approach often employed of focusing on a single case study. This topic is important as the representation of microphysics is a major source of uncertainty in NWP models. The paper is reasonably well written and the conclusions seem to be robust. My main concerns on the science center on 1) consistency between the radar forward simulator and internal assumptions and characteristics within the microphysics schemes, and 2) inconsistencies in the analysis of P3 (primarily for graupel/hail). These concerns are detailed in major comments below. I also include several minor comments, mainly to clarify certain issues or improve the presentation. Finally, a handful of technical/editorial comments are included at the end. Overall, I think the paper could be acceptable if the main comments are addressed.

**Recommendation:** *Major revision*

We would like to thank the reviewer for this review, which helped to improve the quality of our manuscript. Our point-by-point response is highlighted in blue below. The changes made to the manuscript are highlighted in red. A manuscript version in which all changes are highlighted is provided along with the revised manuscript.

**Major comments.**

1. One of my main comments concerns the consistency of assumptions and ice properties internal to the microphysics schemes with assumptions in the radar forward operator calculations using CR-SIM. It's mentioned a few times that these calculations are consistent, but in my view more detail is needed on this. For instance, what ice particle properties are needed or assumed by the forward operator calculations? I guess this includes particle density, size distribution information, and particle habit? Are there other inputs or assumptions made about ice particles in CR-SIM? I think it should be straightforward to couple the traditional bulk schemes (Thompson, Morrison) with CR-SIM. However, this seems less straightforward for coupling with SBM and particularly P3. In P3, the particle density (e.g., particle mass-size relation) varies across different regions of the size distribution. How was this accounted for? Particularly relevant for this paper, in all the schemes what is assumed for the distribution of liquid water on melting ice particles? This is particularly important in this paper given the focus on radar signatures of hail and particularly the reflectivity bias in all schemes in conditions of large hail (i.e., high reflectivities).

Perhaps these issues are discussed in some of the previous papers on CR-SIM, but more discussion is needed in this paper.

Many of the assumptions of the radar forward operator were discussed in Köcher et al. (2022) in their Section 2.4. The information about the CR-SIM forward operator comes mostly from their publication (Oue et al., 2020) and to a small extent from the study of their Fortran code (which is freely available on the Stonybrook University website: https://you.stonybrook.edu/radar/research/radar-simulators/). However, we are aware that the forward operator is very important for this topic. Therefore, we have extended the part about the forward operator in this paper and now directly provide more information about the assumptions of the forward operator, as well as the relevant citations for more details:

Section 2.1:

> Key assumptions of CR-SIM include particle shapes and particle orientations: cloud droplets are assumed to be spherical, and raindrops and graupel are assumed to be oblate spheroids with aspect ratios that depend on droplet size according to Brandes et al. (2002) and Ryzhkov et al. (2011), respectively. Snow and cloud ice are assumed to be oblate with fixed aspect ratios of 0.6 and 0.2, respectively. P3 deviates from the traditional schemes regarding ice. Here, CR-SIM assumes that small ice and graupel are spherical, while unrimed and partially rimed ice is assumed to be oblate with an aspect ratio of 0.6. In terms of particle orientation, CR-SIM assumes that all particles are 2D Gaussian-distributed with zero mean canting angle according to Ryzhkov et al. (2011). The width of the angle distributions varies depending on the hydrometer class: 10° for clouds, rain, and ice and 40° for snow, unrimed ice, partially rimed ice, and graupel. In terms of particle densities and particle size distributions, CR-SIM is consistent with the applied microphysics schemes. In most of the applied schemes, the particle density is constant and varies only between hydrometeor classes. Only the P3 scheme deviates from this, where the particle density of ice is not constant, but several mass-size relations are used. Following the P3 scheme, CR-SIM also uses multiple mass-size relations. In terms of particle size distributions, CR-SIM follows the same gamma distributions as the bulk microphysics schemes. The spectral bin scheme is a little more complicated. Here CR-SIM requires an additional input file containing the actual bins as simulated by the SBM scheme. There is no melting scheme applied by CR-SIM and as a result, radar signatures resulting from mixed-phased particles, such as for example a "bright band" (Austin and Bemis, 1950) cannot be simulated by the model. For more details and a discussion on the assumptions made by the radar forward operator CR-SIM, refer to (Köcher et al., 2022, Sect 2.4).

Regarding melting particles: The CR-SIM forward operator does not apply a melting scheme. Particles are either completely frozen, or completely melted, in accordance to the corresponding microphysics scheme. This of course means that the model will not be able to reproduce radar signatures produced by melting particles, which is not acknowledged so far. We added a part to the discussion acknowledging the fact that the absence of melting particles (especially melting hail) might add to the lack of simulated reflectivities above 50 dBZ:

Section 5.1:

> (3) Melting particles: The microphysics schemes applied do not consider particles that are partially melted, all particles are either completely frozen or completely melted. The radar forward operator CR-SIM does not apply a melting scheme either. That means, it is not only impossible to reproduce certain radar signatures related to melting (e.g., a "bright band"), but also the increase in reflectivity due to partially melted hail particles cannot be simulated. The highest reflectivity events observed could be due to partially melted hail particles, which would translate to an increase in reflectivity that is not reproducible by the models applied in this study.

Section 6:

This might be related 1) to limitations due to resolution, 2) to density assumptions that are not representative for high density hail-like particles or 3) the absence of partially melted particles in the simulations.

2. Figure 5 and ~p. 16. I think the analysis here is a misinterpretation of graupel and rimed ice as simulated by P3. Graupel-like or even hail-like ice particles in P3 don't necessarily consist wholly of rimed ice mass, -- they may contain substantial mass grown by vapor deposition as well (but still have high particle densities expected for graupel/hail). Thus, I think it's incorrect to only include rimed mass for the mixing ratio threshold in Fig. 5 for P3. If the hydrometeor ID identifies a particular time/location as hail/graupel from the P3 output, I would include all ice (rime plus vapor grown) in the analysis. This is particularly relevant to the analysis of graupel/hail since there is no separate rimed ice category in P3. It seems quite likely that ice present at the 1 km level in P3 for summer convective cases will be fast falling and with a high rime fraction (and therefore hail-like) anyway.

Thank you for this comment. We indeed misunderstood graupel as it is simulated by P3. In their paper, Morrison and Milbrandt (2015) state that graupel is "filled in with rime", which we misinterpreted as consisting of only rimed mass. This is, as we understand now, not the meaning of this sentence. This is a very important comment, as it changes the graupel analysis regarding the P3 scheme. As suggested, we include now all ice in the P3 analysis. For a fair comparison, we expanded this to the other schemes as well and now include all ice classes for all schemes and the hydrometeor classification. Most of the corresponding figure (Fig. 5) did not change much, which we think confirms the statement that at 1 km altitude during our convective summer cases, most of the ice is graupel/hail-like anyways. However, some changes are visible, which means that there is also other ice present, even though the majority is made of graupel/hail. Noteworthy is probably the area of 'ice' events at 1 km height, based on mixing ratio (center right image of Figure 5, see the updated figure below). Relative to the other schemes, the P3 scheme now simulates less area of 'ice' events, which means that in the P3 scheme, the fraction of graupel/hail-like ice on the total ice in 1 km is a little higher than in the other schemes. However, our general interpretation of the results is not affected by this, because the area of 'graupel/hail' events in the P3 scheme based on mixing ratio was at the lower end of the schemes prior to these changes anyways. For a complete picture, we provide the hail/graupel figure (where really only hail/graupel classes are included, hence without the P3 scheme) in the supplement.

Please find the updated figure below (Note that we also adjusted the y-axis scale, to be consistent with the changes to the PSD-figure from comment 4).

[Figure]

*Figure 5: Same as Fig. 3, but for ice statistics. Frequency (left column) and area (right column) of ice events above various thresholds. Minimum (dashed-dotted line), mean (solid line), and maximum (dashed line) over the 30-day data set. First row: statistics based on simulated and observed reflectivity thresholds at 1 km altitude. Simulated reflectivity from WRF model output after applying the CR-SIM forward simulator (Oue et al., 2020). Second row: statistics based on mixing ratio thresholds at 1 km altitude. Third row: statistics based on thresholds for mixing ratio at the surface. The y-axis on the right side is logarithmically scaled, except for a small range around zero (0-1) with a linear scale.*

In the paper, we changed the text at many instances in order to comply with the changes regarding the graupel/hail (now general ice) analysis. All instances of "graupel/hail" analysis have been renamed to "ice" analysis, except at instances where the conclusions really only concern graupel/hail. This concerns section 1, 2.2, 5 and 6.
Please refer to the marked up version that is provided along with this response for the details.

The wording in Section 5 has been changed more thoroughly. This concerns the following text passages:

Section 2.2:
Of all the ice classes, technically only the hail and graupel classes are of interest for high-impact weather. However, the microphysics schemes applied in this study do not provide exactly the same ice classes. For a fair comparison, we therefore include all ice classes into a common ice category. We assume that all these classes almost exclusively occur related to hail/graupel events during our summer time study period. Thus, vertical ice, wet snow, aggregates, ice crystals, high density graupel, low density graupel, hail, melting hail/big drops are combined and referred to as "ice".

Section 5:
The P3 scheme does not provide a hail or graupel class. Therefore, to allow for a fair comparison between the schemes, we included all ice into this analysis. Given that the dataset consists of convective cases mainly in summer, most of the ice present at 1 km altitude and below is graupel or hail-like anyways. For completeness, the analysis restricted to graupel and hail only (and thus without the P3 scheme) is provided in the supplement.

Section 5.2:
The middle row of Fig. 5 shows the same analysis as the top row, this time for the model's initial ice mixing ratios. A clear ranking can be seen between the microphysical schemes, both for area and for the frequency of ice events, which is almost independent of the mixing ratio threshold: the spectral bin simulations produced the most frequent and widespread ice events on average, while the P3 produced ice events that were the smallest and least frequent. The majority of the ice events at the height of 1 km are graupel events, because slower falling ice, like aggregates or cloud ice, is melting before it reaches the 1 km altitude.

3. I didn't see any discussion on the scale of the radar observations versus the model, other than discussion of 400 m grid spacing possibly explaining the low bias for the instances of high reflectivity associated with graupel/hail. Is there a general scale mismatch between the radar and the observations? Any scale mismatch is likely to be quite important for instances of heavy rain and hail. If the radar data is higher resolution, can it be appropriately averaged to give a comparable scale to the model data? Keep in mind that the effective resolution in models like WRF is about 5-7 times the grid spacing (Skamarock 2004, MWR).

The DWD radar data contain bins every 250 m along the range axis and every 1° along the azimuth axis. This corresponds to a distance between two azimuth radar bins of about 265 m at the nearer edge of the domain, about 685 m in the center of the domain, and about 1100 m at the far edge. The radar volume data is provided with 11 elevation angles

from 0.5° to 25° elevation, so there are less vertical levels in the radar compared to the model. The radar resolution changes with distance to the radar, which makes a scale analysis difficult. However, a radar also requires more than two samples to resolve an event, according to the Nyquist theorem. With a nominal grid sampling of about 700 m at the domain center, the effective resolution at the domain center is at a similar magnitude to the models effective resolution. However, we understand that this must be discussed, so we extended the data and method section with a part that states these numbers and the underlying assumptions on the resolution.

Section 2.1:

*The horizontal model grid spacing is at 400 m. The DWD radar data is provided with bins every 250 m along the range axis and every 1° along the azimuth. The beamwidth is about 265 m at the closer domain edge, about 685 m at the domain center, and about 1100 m at the far edge. Simulated and measured radar signals are converted to a regular Cartesian grid with a grid spacing of 400 m using inverse range interpolation. This interpolation includes the four nearest data points, weighted by their distance $(1/distance)^2$ . Both, radar and model, require a sufficient spatial sampling to observe a physical phenomenon like a strong precipitation cell. Here, this translates into the question of an effective resolution, which is coarser than the nominal resolution. Skamarock (2004) estimates the effective resolution of WRF to be 5-7 times the nominal resolution, which would result in around 2 km effective model resolution in our case. Given that a radar with a nominal sampling of around 700 m (at the domain center) also needs at least 2-3 samples of a precipitation cell to begin resolving its true intensity, the "effective resolution" of such observations seems to be in a comparable range.*

4. The SBM results from Figure 4 look rather strange, in particular, the sharp drop at diameters just above 4 mm. What is the explanation for this? Does it have something to do with how breakup is treated in this scheme? Or is this the maximum size of bins in the scheme?

The sharp drop means that no droplets larger than ~4 mm were simulated with the SBM scheme. The ~4-mm bin is the third highest bin, so in principle it should be possible to produce larger drops in the SBM simulations. However, the rain mass in the two largest bins is almost always 0 (or very close to 0) during our 30-d simulations. According to Shpund et al. (2019), a drop breakup scheme is applied that follows Kamra et al. (1991) and Srivastava (1971) and includes spontaneous breakup and collisional breakup. A snow breakup scheme is also applied, implying that raindrops formed from melting snow could be limited in size due to snow breakup. We believe it is possible that drop-breakup prevents the formation of the largest droplets. In any case, we believe that our current figure visually enhances the drop to 0 because of the logarithmic y-axis scale that never reaches 0. The large x-axis with raindrops up to 9 mm in size likely further enhances this impression. We therefore cut the x-axis at 6.6 mm (corresponding to the largest bin in the SBM model) and extended the y-axis to 0. This reveals that the simulated rain number concentration is 0 starting at the second laragest bin (~5 mm).

[Figure]

*Figure 4: Mean rain drop size distribution over all 30 days at 1 km altitude. The y-axis is logarithmically scaled, except for a small range around zero (0-1) with a linear scale.*

In the paper, we have added the relevant information about the break up schemes and the largest SBM bin:

Section 4.3:

The opposite is true for large raindrops greater than 4 mm, which were not simulated at all by the SBM scheme. In principle, the SBM can simulate larger raindrops, as the largest bin corresponds to raindrops of 6.6 mm diameter. According to Shpund et al. (2019), there is a drop break-up scheme applied that follows Kamra et al. (1991) and Srivastava (1971), which includes spontaneous breakup and collisional breakup. There is also a snow breakup scheme applied, which might limit the rain drop sizes for rain created from melting snow.

**Minor comments.**

Abstract, line 7. "heavy rain events" is imprecise. Can you state here the specific rain rates used to define such events?
The used rain rates are now specified:

*All simulations, regardless of the microphysical scheme, predict heavy rain events **(15, 25, and 40 mm per hour)** that cover larger average areas than those observed by radar.*

Line 26. I disagree that increasing resolution eliminates problems caused by inaccurate parameterizations, it addresses some of these problems. Thus, I suggest adding "some" before "inaccurate parameterizations".
What we mean by that sentence is that an increase in resolution to the point that a process can explicitly calculated makes a parameterization unnecessary and hence removes any problem introduced by that parameterization. However, we understand that it is realistic only for some parameterizations to be removed by an increase in resolution. Therefore, we added the word "some" as suggested.

Line 32. Add "usually" before "predict", since not all bulk schemes assume a predefined statistical function for the particle size or mass distributions (a few schemes predict processes directly from moments without assuming any functional form of the size distribution, e.g. Kogan and Belochitski 2012, JAS).
Changed as suggested.

Line 41. Not sure I agree that it is not known exactly which processes are poorly represented and I don't think the paper cited here (Morrison et al. 2020) makes this argument either – we have some idea of which processes are most uncertain or most poorly represented. Perhaps reword this sentence to "It is known that many processes, especially those involving ice microphysics, are poorly represented in numerical weather prediction models (Morrison et al., 2020)."
Changed as suggested.

Line 151. I'd add "single" before "case study", as I think this states the point here better.
Changed as suggested.

Line 184. Not clear what you mean by "The missing information about the area of rain events is presented in the top right of Fig. 3." What is missing? I think what you mean is simply "The area of rain events is presented in the top right of Fig. 3."?
Changed as suggested.

Line 188. Confusing as written. Suggest removing "number of".
Changed as suggested.

Line 240. Are the differences really "astounding"? Maybe "substantial" ort "major" would be better?
Changed to "substantial".

Lines 244-246. You might note that these relations of mass and reflectivity to diameter are true for liquid drops (or more generally, isometric particles).
Changed as suggested.
> *For (isometric) liquid drops, mass mixing ratio depends to the third power ($\propto D^3$) on particle diameter, while reflectivity depends to the sixth power ($\propto D^6$) on particle diameter.*

Line 247. Technically, all of the bulk schemes here use complete size distributions, meaning they extend mathematically from 0 to infinity. Thus, it's better to just say "..few large particles may contribute significantly..." rather than "few or no large particles...".
Changed as suggested.

Line 268. I think this could be reworded more clearly – I suggest replacing "due to the missing large raindrops" with "due to the lack of large raindrops".
Changed as suggested.

Line 290. Technically droplet fall velocity and droplet size distribution are not processes. Suggest rewording this to "is affected by processes such as evaporation and drop sedimentation, ..."
Changed as suggested.

Line 300. They don't evaporate faster because of high surface tension, it's because evaporation in the schemes depends mainly on the number concentration times the mean radius (referred to as the integral radius), with some additional modification to account for ventilation.
Thank you for this correction. We removed this sentence completely, as we think the exact details of why small droplets evaporate faster should not be in the focus and distract from the actual point made.

Lines 315-316. This is not correct – the P3 scheme simulates not just number and mass mixing ratios of ice. By predicting additional ice attributes, it can distinguish between graupel-like and hail-like ice (for example, by differences in mean density, size, and fallspeed).
Thank you for this correction. In the context of the major comment 2, we include now all ice in the analysis, which is why this sentence was obsolete anyways and exchanged with the following two sentences:
> The P3 scheme does not provide a hail or graupel class. Therefore, to allow for a fair comparison between the schemes, we included all ice into this analysis.

Lines 331-332. This is confusing. I'd reword to "a grid spacing less than about 250 m is required...".
Changed as suggested.

Line 333. I'd add "when" before "further".
Changed as suggested.

Line 339. Note that a multiple "free" category version of P3 exists (see Milbrandt and Morrison 2016, JAS). Thus, I'd reword this to "...this version of the P3 scheme uses only one ice class...".
Changed as suggested.

Line 378. "likely" seems too strong of a word to use here. Perhaps reword to "which might be a resolution problem". Same comment on line 415.
We included more possible reasons for the missing extreme reflectivites. The sentence reads now:
> In summary, for the ice statistics, no model is able to reproduce the most extreme reflectivity statistics of greater than 55 dBZ, which might be a problem with density assumptions, the absence of partially melted particles in the simulations or a resolution issue.

**Technical/editorial comments.**

Line 4, abstract. Suggest adding "the" before "observation dataset".
Changed as suggested.

Line 44. Perhaps replace "just this" with "such".
Changed as suggested.

Line 78. Replace "are" with "were".
Changed as suggested.

Line 84. Add "schemes" after "microphysics"?
Changed as suggested.

Line 99. The first "is" should be "are" (data here is plural).
Changed as suggested.

Line 179. I feel "then" could be removed.
Changed as suggested.

Line 226. Replace "it is relying on" with "it relies on".
Changed as suggested.

Line 227. "mixing ratio" should be "mixing ratios".
Changed as suggested.

Line 260. I think "github" should be "GitHub"?
Changed as suggested.

Line 260. Reword to "Because the bulk schemes do not actually have fixed size bins..."
Changed as suggested:
> *Because the bulk schemes do not actually have fixed size bins, the number concentration can be calculated for any droplet size.*

Line 272. "simulate" should be "simulates". Alsop "produce" should be "produces". Same comment on the next line (line 273) as well.
Changed as suggested:
> *But given that the SBM scheme simulates high mixing ratio of rain mass, but at the same time produces too few heavy rain events based on the reflectivity produced, it stands to reason that this scheme generally produces too few large raindrops.*

Line 278. "is" should be "was" and "note" should be "noted".
Changed as suggested.

Line 279. "attribute" should be "attributed".
Changed as suggested.

Lines 277-281. This sentence is very long, perhaps break it up into 2 sentences.
Split up to two sentences:

*They noted that none of their simulations were able to successfully reproduce the observed polarimetric radar signatures. This is attributed to median raindrop sizes that are too large (Morrison and Thompson schemes) and a simulated frequency of very large raindrops lower than observed (Thompson scheme).*

Lines 281-286. I'd suggest using past tense in the writing here, since you're describing what previous studies found.
Switched to past tense as suggested (changes in bold):

*In contrast, Putnam et al. (2016) **found** that both Morrison and Thompson 2-mom produce reflectivity values that are too high, which they **attributed** to PSDs containing too many large drops, too much precipitation coverage, and, in the case of the Morrison simulations, a bias due to wet graupel. With respect to our study, we can confirm too much precipitation coverage, and our results suggest that there are too many large raindrops in the Thompson simulations, which is consistent with Putnam et al. (2016) but in contrast to Wu et al. (2021). However, both studies **evaluated** the microphysical schemes using only case studies, which is not generally applicable to different weather situations.*

Line 297. Add a comma after "schemes". Also, replace "is getting smaller" with "becomes smaller".
Changed as suggested.

Line 298. "schemes" should be "scheme".
Changed as suggested.

Line 301. I think you can remove "also".
Changed as suggested.

Line 312. Add "are" before "of interest".
Changed as suggested.

Line 331. There's an extra right parenthesis after "(2015)".
Removed the extra parenthesis.

Line 332. Space is missing between "area" and "converge".
Added a space between "area" and "coverage".

Line 333. Remove "with".
Changed as suggested.

Line 335. I think "reflection" should be "reflectivity"?
Changed as suggested.

Line 338. I'd replace "from" with "with".
Changed as suggested.

Line 343. Replace "distribution" with "distributions".
Changed as suggested.

Line 374. Typo: "is" is repeated twice.
Removed one "is".

Figure 3 caption. "Oue et al. (2020)" should be "(Oue et al., 2020)". Same comment with the Figure 5 caption as well.
Changed as suggested.

**References.**

Kogan, Y. L., and A. Belochitski, 2012: Parameterization of cloud microphysics based on full integral moments. J. Atmos. Sci., 69, 2229-2242.

Milbrandt, J. A., and H. Morrison, 2016: Parameterization of cloud microphysics based on the prediction of ice particle properties. Part 3: Introduction of multiple free categories. J. Atmos. Sci., 73, 975-995.

Skamarock, W. C., 2004: Evaluating mesoscale NWP models using kinetic energy spectra. Mon. Wea. Rev., 132, 3019-3032.

**References**

Austin, P. M. and Bemis, A. C.: A quantitative study of the "bright band" in radar precipitation echoes, J. Atmos. Sci., 7, 145–151, 1950.

Brandes, E. A., Zhang, G., and Vivekanandan, J.: Experiments in rainfall estimation with a polarimetric radar in a subtropical environment, J. Appl. Meteorol., 41, 674–685, 2002.

Kamra, A. K., Bhalwankar, R. V., & Sathe, A. B. (1991). Spontaneous breakup of charged and uncharged water drops freely suspended in a wind tunnel. Journal of Geophysical Research – Atmospheres, 96(D9), 17159–17168. https://doi.org/10.1029/91jd01475

Köcher, G., Zinner, T., Knote, C., Tetoni, E., Ewald, F., and Hagen, M.: Evaluation of convective cloud microphysics in numerical weather prediction models with dual-wavelength polarimetric radar observations: methods and examples, Atmos. Meas. Tech., 15, 1033–1054, https://doi.org/10.5194/amt-15-1033-2022, 2022.

Morrison, H. and Milbrandt, J.A.: Parameterization of cloud microphysics based on the prediction of bulk ice particle properties. Part I: Scheme description and idealized tests, J. Atmos. Sci., 72, 287–311, 2015.

Oue, M., Tatarevic, A., Kollias, P., Wang, D., Yu, K., and Vogelmann, A. M.: The Cloud-resolving model Radar SIMulator (CR-SIM) Version 3.3: description and applications of a virtual observatory, Geosci. Model Dev., 13, 1975–1998, https://doi.org/10.5194/gmd-13-1975-2020, 2020.

Putnam, B. J., Xue, M., Jung, Y., Zhang, G., and Kong, F.: Simulation of polarimetric radar variables from 2013 CAPS spring experiment storm-scale ensemble forecasts and evaluation of microphysics schemes, Mon. Weather Rev., 145, 49–73, 2017.

Ryzhkov, A., Pinsky, M., Pokrovsky, A., and Khain, A.: Polarimetric radar observation

operator for a cloud model with spectral microphysics, J. Appl. Meteorol. Clim., 50, 873–894, 2011.

Shpund, J., Khain, A., Lynn, B., Fan, J., Han, B., Ryzhkov, A., Snyder, J., Dudhia, J., and Gill, D.: Simulating a Mesoscale Convective System Using WRF With a New Spectral Bin Microphysics: 1: Hail vs Graupel, J. Geophys. Res.-Atmos., 124, 14072–14101, 2019

Skamarock, W. C.: Evaluating Mesoscale NWP Models Using Kinetic Energy Spectra, Monthly Weather Review, 132, 3019–3032, https://doi.org/10.1175/mwr2830.1, 2004.

Srivastava, R. C. (1971). Size distribution of raindrops generated by their breakup and coalescence. Journal of the Atmospheric Sciences, 28(3), 410–415. https://doi.org/10.1175/1520-0469(1971)028<0410:SDORGB>2.0.CO;2

Wu, D., Zhang, F., Chen, X., Ryzhkov, A., Zhao, K., Kumjian, M. R., Chen, X., and Chan, P.-W.: Evaluation of Microphysics Schemes in Tropical Cyclones Using Polarimetric Radar Observations: Convective Precipitation in an Outer Rainband, Monthly Weather Review,595 149, 1055–1068, https://doi.org/10.1175/mwr-d-19-0378.1, 2021.

---

## Referee Report (RR1)

Second review of "**Influence of cloud microphysics schemes on weather model predictions of heavy precipitation**", by Kocher et al., submitted to *ACP*.

**General comments.** The paper is improved from the previous version. The authors have well addressed my main concerns with the previous version, particularly consistency of CR-SIM with the microphysics scheme assumptions and consistency of the ice analysis with how P3 represents ice particle properties. I have several additional comments and suggestions below to mainly improve the presentation; all are minor. Once these are addressed I recommend accepting the paper.

NOTE: line numbers refer to track changed version.

**Overall recommendation:** *Minor revision*

**Minor science and editorial comments.**

1. General (incl. abstract and main text): terminology of "model" versus "scheme". There seems to be a mix and match of this (e.g., lines 7 and 10 in abstract says "scheme", but then lines 12, 13, 14 say "model"). Usually "model" refers to the main model system or driver model, like WRF or ICON. I suggest using consistent terminology throughout the paper where "scheme" refers to the microphysics parameterization and "model" refers to the main driver model or modeling system.

2. Which version of P3 was used (there are multiple versions available in WRF)? Was it the two-moment, single category version (MP option 50)? It would be good to clarify this somewhere in the paper. Perhaps even give the MP option #'s for all the schemes used here, to make it clear to readers for all the schemes tested.

3. Lines 86-87. (1) and (2) are not approaches per se; they are questions covering specific science problems. Thus I suggest rewording by not calling (1) and (2) "approaches" or clarifying these are approaches to address questions (1) and (2).

4. Line 110. "particle property prediction" should be "Predicted Particle Properties" (and it is typically capitalized).

5. p. 4-5, this paragraph is very long. It might help to improve the structure by starting a new paragraph beginning with the sentence "A forward radar operator (CR-SIM; Oue et al., 2020)…" Also, I'd start another new paragraph beginning with the sentence "The horizontal model grid spacing is at 400 m." since this is covering a different topic than CR-SIM discussed in previous sentences.

6. Lines 117-120. Here you refer to different ice types in P3 (e.g. small ice, graupel, unrimed and partially rimed ice). Do this mean the different ice types distributed across the particle size distribution (PSD) as in Fig. 1 of Morrison and Milbrandt (2015)? If so, I'd clarify this and

perhaps cite Fig. 1 in Morrison and Milbrandt (2015). Is CR-SIM coupled in a consistent way with these different regions of the PSD?

7. Line 135. Suggest replacing "Both, radar and model, require…" with "Both the radar and model require…".

8. Line 160. Would "quantities" be better than "moments"? Are these quantities formally moments of a distribution? (I don't think so).

9. Line 184-185. Confusing wording here. Suggest replacing "and P3, the simulations are for the most part even able to…" with "and P3 are for the most part even able to…".

10. Line 190. "relevant over a larger statistic" is confusing. Maybe replace with "relevant statistically over a longer period".

11. Lines 195-200. Same comment as #3 above referring to (1) and (2) as approaches. Can this be reworded?

12. Line 202. Typo: "cays" should be "days".

13. Line 231-237. There are several places here that report rain rates in units of $1/m^2$. I don't understand this. Is it a typo or error? Same comment in the Fig. 3 caption as well. If the units are indeed $1/m^2$ can you explain where that comes from?

14. Line 238, I don't follow this sentence. Can it be reworded or clarified?

15. Line 275. In my previous review I had suggested that the authors clarify that the $D^3$ and $D^6$ relations for mass and reflectivity apply to liquid drops. However, whether this requires isometric particles depends on how diameter is actually defined. For larger liquid drops which of course are not isometric (since their aspect ratio changes with size), this problem is resolved by defining the diameter as that of a volume-equivalent sphere. All this to say, I'd simply remove "(isometric)" from this sentence as that might be confusing.

16. Line 300. Maybe "behaviors" instead of "behavior"?

17. Line 314. "is" should be "was" for consistency of tense.

18. Line 333. Suggest replacing "is undergoing" with "undergoes".

19. Line 335. Add "rate" after "evaporation".

20. Line 336. Replace "in" with "at".

21. Line 337. Add "and" before "all schemes".

22. Line 340. Replace the comma after "thresholds" with a semi-colon.

23. Line 342. "indicate" should be "indicates" for subject-verb agreement.

24. 346-347. Suggest rewording this to "However, hail events also have damage potential and therefore are of interest."

25. Line 353. Suggest a small rewording to "The P3 scheme does not have a separate hail or graupel class."

26. Line 354. Suggest replacing "into" with "in"

27. Line 356. "anyways" should be "anyway".

28. Line 360. Suggest replacing "the microphysics schemes" with "microphysics scheme".

29. Line 365. I'd reword this sentence. SBM not producing a single instance of ice at 35 dBZ or higher is one of two possibilities of a binary situation (either this occurs, or not), so it's awkwardly worded in this sentence to say the same is true for the Morrison scheme but to a lesser extent.

30. Line 374. Remove the first comma. Also, remove "when".

31. Line 376. I think you mean "too large" not "too small".

32. Lines 377-378. This may be true with the way the schemes are configured here, but the Morrison scheme in WRF does have an option to use properties of hail (with a high density of 900 kg m$^{-3}$) rather than graupel for the rimed ice category. This could be mentioned here (perhaps in a footnote?).

33. Lines 381-386. A newer version of P3 actually *does* include partially melted ice (Cholette et al. 2019). This is not yet implemented in WRF, but could be mentioned here.

Cholette, M., H. Morrison, J. A. Milbrandt, and J. M. Theriault, 2019: Parameterization of the bulk liquid fraction in the Predicted Particle Properties (P3) scheme: Description and idealized tests. *J. Atmos. Sci.*, 76, 561-582.

FYI there is a paper also just accepted in JAMES that applies this newer version of P3 to simulations of a squall line, and discusses the impact of wet ice on the reflectivity calculation:

Cholette, M., J. A. Milbrandt, H. Morrison, D. Paquin-Ricrad, and D. Jacques, 2022: Combining triple-moment ice with prognostic liquid fraction in the P3 microphysics scheme: Impacts on a simulated squall line. *J. Adv. Mod. Earth Sys.* (accepted)

34. Line 393. Suggest replacing "The P3" with "P3".

35. Line 394. See comment #32 above regarding the graupel/hail switch in the Morrison scheme, which is relevant here as well. Also relevant to the sentence on line 399.

36. Line 395. For consistent tense with the rest of this paragraph, I'd replace "were" with "are".

37. Line 397. "reason" should be "reasons".

38. Line 405. Suggest adding "but" before "this time".

39. Line 407. Since you've generally referred to the spectral bin scheme as SBM in the rest of the paper, my suggest is to replace "the spectral bin simulations" with "the SBM simulations".

40. Line 408. Remove "the".

41. Line 415. Suggest replacing "is melting" with "melts".

42. Line 431. I would replace "indicates" with "suggests" because in principle other processes could be responsible for more rapid decrease of ice toward the surface, like changes in fallspeed leading to divergence. Although I agree that greater melting is the most likely explanation.

43. Line 437. Suggest replacing "the Morrison and SDM" with "Morrison and SDM".

44. Line 472. Replace "summerly" with "summer".

45. Line 479. Add "model" before "resolution".

46. Line 480. Not sure what "Tt" is supposed to be. Maybe "For"?

47. Lines 500-502. This sentence is very long. You might consider breaking it into 2 sentences.

---

## Referee Report (RR2)

General comment

I believe this paper is a very interesting contribution to highlight the value of dual-pol observations for the evaluation of microphysics scheme in convective scale models.

The addition of information related to the forward operator and to the density options in the different schemes is very welcomed in this new version.

The new analysis where all ice species are gathered enables an easier comparison from one scheme to the other.

I suggest publishing the paper as it is (just correcting for remaining typo errors):

l 39:
to answer is: How much complexity => to answer is: how much complexity

l 41: knowledge: It is known => knowledge: It is known

l 131:
model grid spacing is at 400 m => is 400 m

l 148:
for each hydrometer type => for each hydrometeor type

line 153: vertical ice ==> you should keep the Dolan (2013) terminology "vertically aligned ice" at least in the text

line 451 :
Tt slightly smaller reflection => The slightly smaller reflection

---

## Author Response (AR2)

**Author responses**

**Author response 1:**

General comment

I believe this paper is a very interesting contribution to highlight the value of dual-pol observations for the evaluation of microphysics scheme in convective scale models.

The addition of information related to the forward operator and to the density options in the different schemes is very welcomed in this new version.

The new analysis where all ice species are gathered enables an easier comparison from one scheme to the other.

I suggest publishing the paper as it is (just correcting for remaining typo errors):

Thank you for the positive review. Our point-by-point review is again highlighted in blue. A marked up-version showing all changes to the manuscript is provided along with the revised manuscript.

l 39:
to answer is: How much complexity => to answer is: how much complexity
Changed as suggested.

l 41: knowledge: It is known => knowledge: It is known
Changed to "knowledge: it is known".

l 131:
model grid spacing is at 400 m => is 400 m
Changed as suggested.

l 148:
for each hydrometer type => for each hydrometeor type
Changed as suggested. Also corrected in lines 120, 276, and 470.

line 153: vertical ice ==> you should keep the Dolan (2013) terminology "vertically aligned ice" at least in the text
Changed as suggested.

line 451 :
Tt slightly smaller reflection => The slightly smaller reflection
Changed to "For slightly smaller reflectivity thresholds".

**Author response 2:**

Second review of "**Influence of cloud microphysics schemes on weather model predictions of heavy precipitation**", by Kocher et al., submitted to ACP.

**General comments.** The paper is improved from the previous version. The authors have well addressed my main concerns with the previous version, particularly consistency of CR-SIM with the microphysics scheme assumptions and consistency of the ice analysis with how P3 represents ice particle properties. I have several additional comments and suggestions below to mainly improve the presentation; all are minor. Once these are addressed I recommend accepting the paper.

NOTE: line numbers refer to track changed version.

**Overall recommendation:** Minor revision

Thank you for this positive and thorough review. Our point-by-point response is highlighted in blue below. The changes made to the manuscript are highlighted in red. A manuscript version in which all changes are highlighted is provided along with the revised manuscript.

**Minor science and editorial comments.**

1. General (incl. abstract and main text): terminology of "model" versus "scheme". There seems to be a mix and match of this (e.g., lines 7 and 10 in abstract says "scheme", but then lines 12, 13, 14 say "model"). Usually "model" refers to the main model system or driver model, like WRF or ICON. I suggest using consistent terminology throughout the paper where "scheme" refers to the microphysics parameterization and "model" refers to the main driver model or modeling system.
Changed as suggested. With "model", we now refer exclusively to the main NWP model and with "scheme" we refer to the microphysics parameterization. Within this context, we also improved consistency regarding the use of "microphysical scheme" and "microphysics scheme". We now consistently use "microphysics scheme" throughout the manuscript.

2. Which version of P3 was used (there are multiple versions available in WRF)? Was it the two-moment, single category version (MP option 50)? It would be good to clarify this somewhere in the paper. Perhaps even give the MP option #'s for all the schemes used here, to make it clear to readers for all the schemes tested.
Yes, it was the version with the MP option 50. We added a table to the data section that shows all employed microphysics schemes, the abbreviation used throughout the manuscript, the WRF-ID and the corresponding publication.

3. Lines 86-87. (1) and (2) are not approaches per se; they are questions covering specific science problems. Thus I suggest rewording by not calling (1) and (2) "approaches" or clarifying these are approaches to address questions (1) and (2).
We were actually not referring to the two science questions, but to the two possibilities of comparing model output with radar observations: (1) by applying a radar forward operator, i.e, comparing in "radar space" and (2) by retrieving microphysical information from radar signals, i.e., comparing in "model space". This was described in lines 69-70 and called "approach" 1 and 2. We understand that is very confusing, so we changed the phrasing, avoiding the numbers and simply stating directly what was done. The paragraph now reads like this:

In Köcher et al. (2022), cloud microphysics schemes of varying complexity are assessed by a statistical comparison of the observed radar signals with the simulated radar signals from the model output. This study builds on the study of Köcher et al. (2022) and goes one step further by additionally retrieving hydrometeor information from the polarimetric radar observations and comparing it with the simulated hydrometeors.

4. Line 110. "particle property prediction" should be "Predicted Particle Properties" (and it is typically capitalized).
Changed as suggested. Now part of the table, as of comment 2. Also changed in abstract.

5. p. 4-5, this paragraph is very long. It might help to improve the structure by starting a new paragraph beginning with the sentence "A forward radar operator (CR-SIM; Oue et al., 2020)…" Also, I'd start another new paragraph beginning with the sentence "The horizontal model grid spacing is at 400 m." since this is covering a different topic than CR-SIM discussed in previous sentences.
Changed as suggested.

6. Lines 117-120. Here you refer to different ice types in P3 (e.g. small ice, graupel, unrimed and partially rimed ice). Do this mean the different ice types distributed across the particle size distribution (PSD) as in Fig. 1 of Morrison and Milbrandt (2015)? If so, I'd clarify this and perhaps cite Fig. 1 in Morrison and Milbrandt (2015). Is CR-SIM coupled in a consistent way with these different regions of the PSD?
Yes, this is how we understand CR-SIM. CR-SIM claims to be consistent with the supported microphysics schemes (Oue et al., 2020). For the P3 scheme, this means CR-SIM must differentiate between the regions of the PSD, as visualized in Fig. 1 of Morrison and Milbrandt (2015). We have not tested our self if this was implemented correctly. However, from inspecting their code (openly available here: ftp://ftp.radar.bnl.gov/outgoing/moue/crsim/src/crsim-3.33.tar.gz ; within src/crsim_subrs.f90), we can see that for the P3 scheme, they indeed calculate critical sizes to separate different regions of the PSD. We now clarify that we refer to the different ice types distributed across the PSD by referring to Fig. 1 of Morrison and Milbrandt (2015) in the text:

P3 deviates from the traditional schemes regarding ice, and uses different ice types distributed across the particle size distribution (see Fig. 1 in Morrison and Milbrandt, 2015). Here, CR-SIM assumes that small ice and graupel are spherical, while unrimed and partially rimed ice is assumed to be oblate with an aspect ratio of 0.6.

7. Line 135. Suggest replacing "Both, radar and model, require..." with "Both the radar and model require...".
Changed as suggested.

8. Line 160. Would "quantities" be better than "moments"? Are these quantities formally moments of a distribution? (I don't think so).
Changed as suggested.

9. Line 184-185. Confusing wording here. Suggest replacing "and P3, the simulations are for the most part even able to..." with "and P3 are for the most part even able to...".
Changed as suggested.

10. Line 190. "relevant over a larger statistic" is confusing. Maybe replace with "relevant statistically over a longer period".
Changed as suggested.

11. Lines 195-200. Same comment as #3 above referring to (1) and (2) as approaches. Can this be reworded?
We think it is not necessary to number the approaches an it seems to confuse more than it helps. So we removed any instance where "approach #" was used.

12. Line 202. Typo: "cays" should be "days".
Changed as suggested.

13. Line 231-237. There are several places here that report rain rates in units of 1/m² . I don't understand this. Is it a typo or error? Same comment in the Fig. 3 caption as well. If the units are indeed 1/m² can you explain where that comes from?
The unit is l/m² (liters per square meters), which equals "mm". It is not a typo, but you are right that the l and the 1 look very similar in the manuscript. This is the journal template, so we cannot change it. However, technically it is l/m² in one hour – so it should be l/(m²h) or mm/h. We decided to use mm/h to avoid the confusion of liters and 1.

14. Line 238, I don't follow this sentence. Can it be reworded or clarified?
What we meant by that sentence is that by converting the heavy rain thresholds of the DWD to reflectivity thresholds,  this should help to give an orientation where "heavy rain" is in terms of reflectivity. We rephrased the sentence like this:

This gives an indication of the reflectivity thresholds that correspond to heavy rain.

15. Line 275. In my previous review I had suggested that the authors clarify that the $D^3$ and $D^6$ relations for mass and reflectivity apply to liquid drops. However, whether this requires isometric particles depends on how diameter is actually defined. For larger liquid drops which of course are not isometric (since their aspect ratio changes with size), this problem is resolved by defining the diameter as that of a volume-equivalent sphere. All this to say, I'd simply remove "(isometric)" from this sentence as that might be confusing.
Thank you for the clarification. We removed "(isometric)" as suggested.

16. Line 300. Maybe "behaviors" instead of "behavior"?
Changed as suggested.

17. Line 314. "is" should be "was" for consistency of tense.
Changed as suggested.

18. Line 333. Suggest replacing "is undergoing" with "undergoes".
Changed as suggested.

19. Line 335. Add "rate" after "evaporation".
Changed as suggested.

20. Line 336. Replace "in" with "at".
Changed as suggested.

21. Line 337. Add "and" before "all schemes".

Changed as suggested.

22. Line 340. Replace the comma after "thresholds" with a semi-colon.
Changed as suggested.

23. Line 342. "indicate" should be "indicates" for subject-verb agreement.
Changed as suggested.

24. 346-347. Suggest rewording this to "However, hail events also have damage potential and therefore are of interest."
Changed as suggested.

25. Line 353. Suggest a small rewording to "The P3 scheme does not have a separate hail or graupel class."
Changed as suggested.

26. Line 354. Suggest replacing "into" with "in"
Changed as suggested.

27. Line 356. "anyways" should be "anyway".
Changed as suggested.

28. Line 360. Suggest replacing "the microphysics schemes" with "microphysics scheme".
Changed as suggested.

29. Line 365. I'd reword this sentence. SBM not producing a single instance of ice at 35 dBZ or higher is one of two possibilities of a binary situation (either this occurs, or not), so it's awkwardly worded in this sentence to say the same is true for the Morrison scheme but to a lesser extent.
We rephrased this part:

> The most extreme case is the SBM scheme, which hardly produces any ice events at higher reflectivities. There is not a single time step within the 30 day dataset at which the SBM scheme simulated ice grid cells of 35 dBZ or higher (Top left image in Fig. 5). However, most of the other schemes, and especially the Morrison scheme, also consistently show fewer ice events compared to the observations.

30. Line 374. Remove the first comma. Also, remove "when".
Changed as suggested.

31. Line 376. I think you mean "too large" not "too small".
That is correct. Changed to "too large".

32. Lines 377-378. This may be true with the way the schemes are configured here, but the Morrison scheme in WRF does have an option to use properties of hail (with a high density of 900 kg m$^{-3}$) rather than graupel for the rimed ice category. This could be mentioned here (perhaps in a footnote?).
It is correct that the Morrison scheme has a switch to use hail with a high density instead. The same is true for a new version of the SBM scheme. However, the configuration applied in this study did not utilize these switches. We clarify this in a footnote now:

> With the configuration that was used in this study.

33. Lines 381-386. A newer version of P3 actually *does* include partially melted ice (Cholette et al. 2019). This is not yet implemented in WRF, but could be mentioned here.

Cholette, M., H. Morrison, J. A. Milbrandt, and J. M. Theriault, 2019: Parameterization of the bulk liquid fraction in the Predicted Particle Properties (P3) scheme: Description and idealized tests. *J. Atmos. Sci.*, 76, 561-582.

FYI there is a paper also just accepted in JAMES that applies this newer version of P3 to simulations of a squall line, and discusses the impact of wet ice on the reflectivity calculation:

Cholette, M., J. A. Milbrandt, H. Morrison, D. Paquin-Ricrad, and D. Jacques, 2022: Combining triple-moment ice with prognostic liquid fraction in the P3 microphysics scheme: Impacts on a simulated squall line. *J. Adv. Mod. Earth Sys*. (accepted)

Thank you for this information, it is much appreciated. Regarding our manuscript, we added another footnote mentioning the newer version that includes partially melted ice:

A newer version of the P3 scheme does include partially melted ice: Cholette et al. (2019).

34. Line 393. Suggest replacing "The P3" with "P3".
Changed as suggested.

35. Line 394. See comment #32 above regarding the graupel/hail switch in the Morrison scheme, which is relevant here as well. Also relevant to the sentence on line 399.
In line 394, we added a part stating that this is true only with the configuration that was used for this study:

With the configuration that was used in this study, P3 is also the only scheme that allows ice particles to reach densities up to 900 kg m$^{-3}$, i.e., to simulate hail-like particles.

We think the sentence on line 399 can stay like it is, because at this point it should be clear that the ice density is depending on the configuration (mentioned twice already at this point) and also, this sentence refers specifically to graupel density, not to hail.

36. Line 395. For consistent tense with the rest of this paragraph, I'd replace "were" with "are".
Changed as suggested.

37. Line 397. "reason" should be "reasons".
Changed as suggested.

38. Line 405. Suggest adding "but" before "this time".
Changed as suggested.

39. Line 407. Since you've generally referred to the spectral bin scheme as SBM in the rest of the paper, my suggest is to replace "the spectral bin simulations" with "the SBM simulations".

Changed as suggested.

40. Line 408. Remove "the".
Changed as suggested.

41. Line 415. Suggest replacing "is melting" with "melts".
Changed as suggested.

42. Line 431. I would replace "indicates" with "suggests" because in principle other processes could be responsible for more rapid decrease of ice toward the surface, like changes in fallspeed leading to divergence. Although I agree that greater melting is the most likely explanation.
Changed as suggested.

43. Line 437. Suggest replacing "the Morrison and SDM" with "Morrison and SDM".
Changed to "Morrison and SBM"

44. Line 472. Replace "summerly" with "summer".
Changed as suggested.

45. Line 479. Add "model" before "resolution".
Changed as suggested.

46. Line 480. Not sure what "Tt" is supposed to be. Maybe "For"?
Changed to "For slightly smaller reflectivity thresholds…"

47. Lines 500-502. This sentence is very long. You might consider breaking it into 2 sentences.
We split up the sentence into 2 sentences:

Within a sub-project of the DFG (German Research Foundation) Priority Programme 2115 (PROM, Trömel et al., 2021), an HMC algorithm is currently being developed for this purpose. This algorithm is based on a clustering approach and an algorithm for quantifying the mixing ratio following Grazioli et al. (2015), Besic et al. (2016) and Besic et al. (2018), and aiming to calculate the mixing ratios of hydrometeor classes as well.

**Other changes:**

We added a sentence at the end of the abstract to paraphrase one of our conclusions that was missing in the abstract:

More complex schemes do not generally yield better results, emphasizing the need to first improve the existing microphysical parameterizations with observational constraints that have the potential to infer microphysical parameters.

Figure improvements:
- changed 'spectral bin' legend entries to 'SBM' for consistency
- changed figure format from png to pdf
- increased font sizes (and adjusted label descriptions to fit)
- Increase x-axis limits for the ice reflectivity (Fig. 5) from 35 dBZ to 32 dBZ

**References**

Morrison, H. and Milbrandt, J. A.: Parameterization of Cloud Microphysics Based on the Prediction of Bulk Ice Particle Properties. Part I: Scheme Description and Idealized Tests, Journal of the Atmospheric Sciences, 72, 287–311, https://doi.org/10.1175/jas-d-14-0065.1, 2015.

Oue, M., Tatarevic, A., Kollias, P., Wang, D., Yu, K., and Vogelmann, A. M.: The Cloud-resolving model Radar SIMulator (CR-SIM) Version 3.3: description and applications of a virtual observatory, Geoscientific Model Development, 13, https://doi.org/10.5194/gmd-13-1975-2020, 2020.

---

## Author Response (AR3)

**Author responses**

We would like to thank the reviewers and the editor for reviewing and accepting our paper. In the current version, we have added one more change as described below. In black, you can find the editor comment. Our answer is in blue. Changes made to the manuscript are highlighted in red.

Thank you for addressing the reviewer comments. With regards to the addition to the abstract I requested I am concerned about the writing, and worry it might dilute the impact of the statement. "More complex schemes do not generally yield better results, emphasizing the need to first improve the existing microphysical parameterizations with observational constraints that have the potential to infer microphysical parameters." Can you please clarify what is mean by " emphasizing the need to first improve the existing microphysical parameterizations with observational constraints that have the potential to infer microphysical parameters"? Observations constraints do not "infer". What parameters? Can the statement be made less wordy? And the link between the clauses is unclear as the second clause seems to argue for possibly greater complexity where the first might be instead for simplicity.

We have shortened the abstract and now simply state the conclusions drawn in the main part of the paper:

> More complex schemes do not necessarily lead to better results in the prediction of heavy precipitation.